# A Unified Framework for Information-Theoretic Generalization Bounds

**Yifeng Chu**  **Maxim Raginsky**
{ychu26,maxim}@illinois.edu[*]

## Abstract

This paper presents a general methodology for deriving information-theoretic generalization bounds for learning algorithms. The main technical tool is a probabilistic decorrelation lemma based on a change of measure and a relaxation of Young's inequality in $L_{\psi_p}$ Orlicz spaces. Using the decorrelation lemma in combination with other techniques, such as symmetrization, couplings, and chaining in the space of probability measures, we obtain new upper bounds on the generalization error, both in expectation and in high probability, and recover as special cases many of the existing generalization bounds, including the ones based on mutual information, conditional mutual information, stochastic chaining, and PAC-Bayes inequalities. In addition, the Fernique–Talagrand upper bound on the expected supremum of a subgaussian process emerges as a special case.

## 1 Introduction

The generalization error of a learning algorithm is a useful proxy for evaluating the performance of the learned model on previously unseen data. Formally, it is defined as the expected (absolute) difference between the population risk and the empirical risk of the hypothesis returned by the algorithm. One of the classical methods for estimating the generalization error is via uniform convergence of various empirical processes indexed by the hypothesis class [1, 2]. For example, in the analysis of Empirical Risk Minimization, one can estimate the expected generalization error via Rademacher averages, which can be bounded from above using chaining techniques [3].

However, the bounds based on uniform convergence are often too pessimistic and may even become vacuous when the hypothesis space is extremely large, a typical situation with deep neural net models. For this reason, it is preferable to obtain algorithm-dependent generalization bounds that take into account the joint distribution of the training samples and of the output hypothesis. In this context, one capitalizes on the intuition that the generalization ability of a learning algorithm should be related to the amount of information the output hypothesis reveals about the training data. This idea, which has origins in the work on PAC-Bayes methods [4, 5], is the basis of the growing literature on information-theoretic generalization bounds, first proposed in [6] and further devoloped in [7–16] and many other works.

In fact, it is possible to effectively combine the information-theoretic approach with the classical framework based on various measures of complexity of the hypothesis class: One can use chaining techniques to successively approximate the hypothesis class by simpler model classes, which can then be analyzed using information-theoretic tools. This methodology, again originating in the PAC-Bayes literature [17], has been developed recently in [18–21]. Our goal in this work is to develop these ideas further by giving a unified framework for information-theoretic generalization bounds, from which many of the existing results emerge as special cases.

---

[*]Department of Electrical and Computer Engineering and Coordinated Science Laboratory, University of Illinois, Urbana, IL 61801, USA.

37th Conference on Neural Information Processing Systems (NeurIPS 2023).

## 1.1 The main idea, informally

The main idea behind our framework is surprisingly simple. We first give an abstract description and then show how it can be particularized to various settings of interest. Let $(X_t)_{t \in T}$ be a centered (zero-mean) stochastic process defined on a probability space $(\Omega, \mathcal{A}, \mathbb{P})$ and indexed by the elements of some set $T$. Let $Q$ be a *Markov kernel* from $\Omega$ to $T$, i.e., a measurable mapping taking each $\omega \in \Omega$ to a probability measure $Q(\cdot|\omega)$ on $T$. Together, $\mathbb{P}$ and $Q$ define a probability measure $\mathbb{P} \otimes Q$ on the product space $\Omega \times T$. The mathematical object we would like to study is the expected value

$$\langle \mathbb{P} \otimes Q, X \rangle := \int_{\Omega \times T} X_t(\omega) Q(\mathrm{d}t|\omega) . \mathbb{P}(\mathrm{d}\omega).$$

For example, assuming that there exists a measurable map $\tau^* : \Omega \to T$, such that

$$X_{\tau^*(\omega)}(\omega) = \sup_{t \in T} X_t(\omega), \qquad \mathbb{P} - \text{a.s.} \tag{1}$$

we can take $Q(A|\omega) := \mathbf{1}_{\{\tau^*(\omega) \in A\}}$ for all measurable subsets $A$ of $T$. Then

$$\langle \mathbb{P} \otimes Q, X \rangle = \mathbf{E}\Big[\sup_{t \in T} X_t\Big]$$

is the expected supremum of $X_t$, the central object of study in the theory of generic chaining, where $(T, d)$ is a metric space and increments $X_u - X_v$ are "stochastically small" relative to $d(u, v)$. Alternatively, consider a statistical learning problem with instance space $\mathcal{Z}$, hypothesis space $\mathcal{W}$, and loss function $\ell : \mathcal{W} \times \mathcal{Z} \to \mathbb{R}_+$. Let $P_Z$ be the (unknown) probability law of the problem instances in $\mathcal{Z}$. Then we could take $\Omega = \mathcal{Z}^n$, $\mathbb{P} = P_Z^{\otimes n}$, $T = \mathcal{W}$, and

$$X_w = \frac{1}{n} \sum_{i=1}^n \big(L(w) - \ell(w, Z_i)\big),$$

where $L(w) := \mathbf{E}_{Z \sim P_Z}[\ell(w, Z)]$ is the *population risk* of $w$. Let $Q$ be a (randomized) learning algorithm that associates to each sample $S = (Z_1, \ldots, Z_n) \sim \mathbb{P}$ a probability measure $Q(\cdot|S)$ on the hypothesis space $\mathcal{W}$. Then

$$\langle \mathbb{P} \otimes Q, X \rangle = \mathbf{E}\Big[\frac{1}{n} \sum_{i=1}^n \big(L(W) - \ell(W, Z_i)\big)\Big]$$

is the expected generalization error of $Q$. In either case, we can proceed to analyze $\langle \mathbb{P} \otimes Q, X \rangle$ via a combination of the following two steps:

- **Decorrelation** — We can remove the correlations encoded in $\mathbb{P} \otimes Q$ by choosing a convenient product measure $\mathbb{P} \otimes \mu$ on $\Omega \times T$, so that (roughly)

$$\langle \mathbb{P} \otimes Q, X \rangle \lesssim \sqrt{D(\mathbb{P} \otimes Q \| \mathbb{P} \otimes \mu)} + \text{Error}$$

  provided the process $(X_t)_{t \in T}$ is regular enough for the error term to be small. Here, we use the relative entropy (or information divergence) $D(\cdot\|\cdot)$ to illustrate the key idea with a minimum of detail; the precise description is given in Section 3.

- **Chaining in the space of measures** — Since the process $(X_t)_{t \in T}$ is centered and $\mathbb{P} \otimes \mu$ is a product measure, we automatically have $\langle \mathbb{P} \otimes \mu, X \rangle = 0$ even though $\langle \mathbb{P} \otimes Q, X \rangle \neq 0$. We can therefore interpolate between $\mathbb{P} \otimes Q$ and $\mathbb{P} \otimes \mu$ along a (possibly infinite) sequence $Q_0, Q_1, \ldots, Q_K$ of Markov kernels, such that $\mathbb{P} \otimes Q_K = \mathbb{P} \otimes Q$, $\mathbb{P} \otimes Q_0 = \mathbb{P} \otimes \mu$, and the differences $\langle \mathbb{P} \otimes Q_k, X \rangle - \langle \mathbb{P} \otimes Q_{k-1}, X \rangle$ are suitably small. Telescoping, we get

$$\langle \mathbb{P} \otimes Q, X \rangle = \sum_{k=1}^K \big(\langle \mathbb{P} \otimes Q_k, X \rangle - \langle \mathbb{P} \otimes Q_{k-1}, X \rangle\big).$$

  For each $k$, we then apply the decorrelation procedure to the *increment process* $(X_u - X_v)_{u,v \in T}$, with $\mathbb{P}$ as before and with a suitably chosen family of couplings of $Q_k(\cdot|\omega)$ and $Q_{k-1}(\cdot|\omega)$. This step can be combined effectively with other techniques, such as symmetrization.

## 2 Preliminaries

**Basic definitions.** All measurable spaces in this paper are assumed to be standard Borel spaces. The set of all Borel probability measures on a space $\mathcal{X}$ will be denoted by $\mathcal{P}(\mathcal{X})$. A *Markov kernel* from $(\mathcal{X}, \mathcal{A})$ to $(\mathcal{Y}, \mathcal{B})$ is a mapping $P_{Y|X} : \mathcal{B} \times \mathcal{X} \to [0, 1]$, such that $P_{Y|X=x}(\cdot) := P_{Y|X}(\cdot|x)$ is an element of $\mathcal{P}(\mathcal{Y})$ for every $x \in \mathcal{X}$ and the map $x \mapsto P_{Y|X=x}(B)$ is measurable for every $B \in \mathcal{B}$. The set of all such Markov kernels will be denoted by $\mathcal{M}(\mathcal{Y}|\mathcal{X})$.

The product of $P_X \in \mathcal{P}(\mathcal{X})$ and $P_{Y|X} \in \mathcal{M}(\mathcal{Y}|\mathcal{X})$ is the probability measure $P_X \otimes P_{Y|X} \in \mathcal{P}(\mathcal{X} \times \mathcal{Y})$ defined on product sets $A \times B \subseteq \mathcal{X} \times \mathcal{Y}$ by $(P_X \otimes P_{Y|X})(A \times B) := \int_A P_{Y|X=x}(B) P_X(\mathrm{d}x)$ and then extended to all Borel subsets of $\mathcal{X} \times \mathcal{Y}$ by countable additivity. This defines a joint probability law for a random element $(X, Y)$ of $\mathcal{X} \times \mathcal{Y}$, so that $P_X$ is the marginal law of $X$, $P_{Y|X}$ is the conditional law of $Y$ given $X$, and $P_Y(\cdot) = \int_{\mathcal{X}} P_{Y|X=x}(\cdot) P_X(\mathrm{d}x)$ is the marginal law of $Y$. The product measure $P_X \otimes P_Y$, under which $X$ and $Y$ are independent, is a special case of this if we interpret $P_Y$ as a trivial Markov kernel with $P_{Y|X=x} = P_Y$ for all $x$.

A *coupling* of $\mu \in \mathcal{P}(\mathcal{X})$ and $\nu \in \mathcal{P}(\mathcal{Y})$ is a probability measure $P \in \mathcal{P}(\mathcal{X} \times \mathcal{Y})$, such that $P(\cdot \times \mathcal{Y}) = \mu(\cdot)$ and $P(\mathcal{X} \times \cdot) = \nu(\cdot)$. We will denote the set of all couplings of $\mu$ and $\nu$ by $\Pi(\mu, \nu)$. Let the space $\mathcal{X} \cup \mathcal{Y}$ be equipped with a metric $d$, and let $\mathcal{P}_p$, for $p \geq 1$, denote the space of all probability measures $\rho$ on $\mathcal{X} \cup \mathcal{Y}$, for which there exists some $z_0 \in \mathcal{X} \cup \mathcal{Y}$ such that $\int_{\mathcal{X} \cup \mathcal{Y}} d(z, z_0)^p \rho(\mathrm{d}z) < \infty$. Then the *p-Wasserstein distance* between $\mu \in \mathcal{P}(\mathcal{X}) \cap \mathcal{P}_p$ and $\nu \in \mathcal{P}(\mathcal{Y}) \cap \mathcal{P}_p$ is given by

$$\mathsf{W}_p(\mu, \nu) := \inf_{\pi \in \Pi(\mu, \nu)} \left( \int d(x, y)^p \pi(\mathrm{d}x, \mathrm{d}y) \right)^{1/p}$$

(see [22, 23] for details).

**$L^p$ and $L_{\psi_p}$ spaces.** The $L^p(\mu)$ norms of $f : \mathcal{X} \to \mathbb{R}$, for $\mu \in \mathcal{P}(\mathcal{X})$ and $p \geq 1$, are defined as

$$\|f\|_{L^p(\mu)} := \left( \int_{\mathcal{X}} |f|^p \, \mathrm{d}\mu \right)^{1/p}$$

whenever the expectation on the right-hand side exists. We will often use the linear functional notation for expectations, i.e., $\langle \mu, f \rangle = \int_{\mathcal{X}} f \, \mathrm{d}\mu$.

For $p \geq 1$, define the function $\psi_p : \mathbb{R}_+ \to \mathbb{R}_+$ by $\psi_p(x) := \exp(x^p) - 1$. Its inverse is given by $\psi_p^{-1}(x) = \left( \log(x+1) \right)^{1/p}$, where $\log$ will always denote natural logarithms. Some useful properties of these two functions are collected in Appendix A of Supplementary Material. The function $\psi_p$ arises in the context of controlling the tail behavior of random variables (see [1, 24, 25] for details). The *Orlicz $\psi_p$-norm* of a real-valued random variable $X$ is defined as

$$\|X\|_{\psi_p} := \inf \left\{ c > 0 : \mathbf{E}\left[\psi_p\left(\frac{|X|}{c}\right)\right] \leq 1 \right\},$$

and the tails of $X$ satisfy $\mathbf{P}[|X| \geq u] \leq K e^{-Cu^p}$ for all $u \geq 0$ and some $K, C > 0$ if and only if $\|X\|_{\psi_p} < \infty$. The Orlicz space $L_{\psi_p}$ is the space of all random variables $X$ with $\|X\|_{\psi_p} < \infty$. In particular, if $X$ is $\sigma$-subgaussian, i.e., $\mathbf{P}[|X| \geq u] \leq 2e^{-u^2/2\sigma^2}$ for all $u \geq 0$, then $\|X\|_{\psi_2} \leq \sqrt{6}\sigma$; conversely, every $X \in L_{\psi_2}$ is $\sigma$-subgaussian with $\sigma \leq c\|X\|_{\psi_2}$ for some absolute constant $c > 0$.

**Information-theoretic quantities.** The relative entropy (or information divergence) $D(\mu\|\nu)$ between two probability measures $\mu, \nu$ on the same space $\mathcal{X}$ is defined as

$$D(\mu\|\nu) := \left\langle \mu, \log \frac{\mathrm{d}\mu}{\mathrm{d}\nu} \right\rangle$$

if $\mu \ll \nu$ (i.e., $\mu$ is absolutely continuous w.r.t. $\nu$), and $D(\mu\|\nu) := +\infty$ otherwise. The following inequality will be useful (proofs of all results are in Appendix B of the Supplementary Material):

**Proposition 1.** *If $\mu \ll \nu$, then for any $p \geq 1$*

$$\left\langle \mu, \psi_p^{-1}\left(\frac{\mathrm{d}\mu}{\mathrm{d}\nu}\right) \right\rangle \leq \left( D(\mu\|\nu) + 1 \right)^{1/p}.$$

The *conditional divergence* between $P_{V|U}, Q_{V|U} \in \mathcal{M}(\mathcal{V}|\mathcal{U})$ given $P_U \in \mathcal{P}(\mathcal{U})$ is defined by

$$D(P_{V|U}\|Q_{V|U}|P_U) := D(P_U \otimes P_{V|U}\|P_U \otimes Q_{V|U}).$$

The mutual information $I(X;Y) := D(P_{Y|X}\|P_Y|P_X)$ and conditional mutual information $I(X;Y|Z) := D(P_{Y|XZ}\|P_{Y|Z}|P_{XZ})$ are special cases of the above definition, and the identities

$$D(P_{Y|X}\|Q_Y|P_X) = I(X;Y) + D(P_Y\|Q_Y) \tag{2}$$
$$D(P_{Y|XZ}\|Q_{Y|Z}|P_{XZ}) = I(X;Y|Z) + D(P_{Y|Z}\|Q_{Y|Z}|P_Z). \tag{3}$$

hold whenever all the quantities are finite. See, e.g., [26] for details.

## 3 The decorrelation lemma

All of our subsequent developments make use of the following *decorrelation lemma*:

**Lemma 1.** *Let $\mu, \nu$ be two probability measures on a space $\mathcal{X}$ such that $\mu \ll \nu$, and let $f, g : \mathcal{X} \to \mathbb{R}_+$ be two nonnegative measurable functions. Then the following inequalities hold:*

$$\langle \mu, fg \rangle \le 2^{1/p}\Big\langle \mu, f\psi_p^{-1}\Big(\frac{\mathrm{d}\mu}{\mathrm{d}\nu}\Big)\Big\rangle + \langle \nu, f\psi_p(g)\rangle \tag{4}$$

*and*

$$\langle \mu, fg \rangle \le 2^{1/p}\|f\|_{L^2(\nu)} + 4^{1/p}\Big\langle \mu, f\psi_p^{-1}\Big(\frac{\mathrm{d}\mu}{\mathrm{d}\nu}\Big)\Big\rangle + 4^{1/p}\|f\|_{L^1(\mu)}\big(\log\langle \nu, \exp(g^p)\rangle\big)^{1/p}. \tag{5}$$

The proof makes extensive use of various properties of $\psi_p$ and $\psi_p^{-1}$. In particular, Eq. (4) is a relaxation of the Young-type inequality $xy \le \psi_p^*(x) + \psi_p(y)$, where $\psi_p^*(x) := \sup_{y \ge 0}(xy - \psi_p(y))$ is the (one-sided) Lengendre–Fenchel conjugate of $\psi_p$. (We refer the reader to [13] for another use of duality in Orlicz spaces in the context of generalization bounds.)

Every use of Lemma 1 in the sequel will be an instance of the following scheme: Let $P_X \in \mathcal{P}(\mathcal{X})$, $Q_Y \in \mathcal{P}(\mathcal{Y})$, and $P_{Y|X} \in \mathcal{M}(\mathcal{Y}|\mathcal{X})$ be given, such that $P_{Y|X=x} \ll Q_Y$ for all $x \in \mathcal{X}$. Let $(X, Y, \bar{Y})$ be a random element of $\mathcal{X} \times \mathcal{Y} \times \mathcal{Y}$ with joint law $P_X \otimes P_{Y|X} \otimes Q_Y$; in particular, $\bar{Y}$ is independent of $(X, Y)$. Furthermore, let $f : \mathcal{Y} \to \mathbb{R}_+$ and $g : \mathcal{X} \times \mathcal{Y} \to \mathbb{R}_+$ be given, such that $\mathbf{E}[\psi_p(g(X, y))] \le 1$ for all $y \in \mathcal{Y}$. Then, applying Lemma 1 conditionally on $X = x$ with $\mu = P_{Y|X=x}, \nu = Q_Y, f$, and $g(x, \cdot)$, and then taking expectations w.r.t. $P_X$, we obtain

$$\mathbf{E}[f(Y)g(X,Y)] \le 2^{1/p}\mathbf{E}\left[f(Y)\psi_p^{-1}\left(\frac{\mathrm{d}P_{Y|X}}{\mathrm{d}Q_Y}(Y)\right)\right] + \mathbf{E}[f(\bar{Y})].$$

In specific cases, the quantity on the right-hand side can be further upper-bounded in terms of the information divergences $D(P_{Y|X}\|Q_Y)$ using Proposition 1.

## 4 Some estimates for the absolute generalization error

We adopt the usual set-up for the analysis of (possibly randomized) learning algorithms and their generalization error. Let an instance space $\mathcal{Z}$, a hypothesis space $\mathcal{W}$, and a nonnegative loss function $\ell : \mathcal{W} \times \mathcal{Z} \to \mathbb{R}_+$ be given. A *learning algorithm* is a Markov kernel $P_{W|S}$ from the product space $\mathcal{Z}^n$ into $\mathcal{W}$, which takes as input an $n$-tuple $S = (Z_1, \dots, Z_n)$ of i.i.d. random elements of $\mathcal{Z}$ with unknown marginal probability law $P_Z$ and generates a random element $W$ of $\mathcal{W}$. We define the *empirical risk* and the *expected* (or *population*) *risk* of each $w \in \mathcal{W}$ by

$$L_n(w) := \langle P_n, \ell(w, \cdot)\rangle = \frac{1}{n}\sum_{i=1}^{n}\ell(w, Z_i), \qquad L(w) := \langle P_Z, \ell(w, \cdot)\rangle = \mathbf{E}[\ell(w, Z)]$$

where $P_n$ is the empirical distribution of $S$, and the *pointwise generalization error* by

$$\mathrm{gen}(w, S) := L(w) - L_n(w).$$

It will also be convenient to introduce an auxiliary $n$-tuple $S' = (Z'_1, \ldots, Z'_n) \sim P_Z^{\otimes n}$, which is independent of $(S, W) \sim P_Z^{\otimes n} \otimes P_{W|S}$. We will use $\tilde{S}$ to denote the pair $(S', S)$ and write $L'_n(w)$ for the empirical risk of $w$ on $S'$.

As a first illustration of our general approach, we show that it can be used to recover some existing results on the generalization error, including the bounds of Xu and Raginsky [7] involving the mutual information and the bounds of Steinke and Zakynthinou [10] involving the conditional mutual infornation. We start with the following estimate on the expected value of $|\text{gen}(W, S)|$:

**Theorem 1.** *Assume the random variables $\ell(w, Z)$, $w \in \mathcal{W}$, are $\sigma$-subgaussian when $Z \sim P_Z$. Let a learning algorithm $P_{W|S}$ be given. Then, for any $Q_W \in \mathcal{P}(\mathcal{W})$,*

$$\mathbf{E}[|\text{gen}(W, S)|] \leq \sqrt{\frac{12\sigma^2}{n}} \left( \mathbf{E}\left[ \psi_2^{-1}\left( \frac{\mathrm{d}P_{W|S}}{\mathrm{d}Q_W} \right) \right] + 1 \right), \tag{6}$$

*where the expectation on both sides is w.r.t. $P_S \otimes P_{W|S} = P_Z^{\otimes n} \otimes P_{W|S}$.*

The key step in the proof is to apply the decorrelation lemma, conditionally on $S$, to $\mu = P_{W|S}$, $\nu = Q_W$, $f(w) = \sigma\sqrt{6/n}$, and $g(w) = \frac{|\text{gen}(w, S)|}{\sigma\sqrt{6/n}}$. The same subgaussianity assumption was also made by Xu and Raginsky [7]. Minimizing the right-hand side of (6) over $Q_W$, we recover their generalization bound up to a multiplicative constant and an extra $O(1/\sqrt{n})$ term (which is unavoidable since we are bounding the expected *absolute* generalization error):

**Corollary 1.** *Under the assumptions of Theorem 1,*

$$\mathbf{E}[|\text{gen}(W, S)|] \leq \sqrt{\frac{24\sigma^2}{n} \left( I(W; S) + 4 \right)}. \tag{7}$$

A notable shortcoming of Theorem 1 and Corollary 1 is that they yield vacuous bounds whenever the mutual information $I(W; S)$ is infinite, which will be the case, e.g., when the marginal probability laws $P_Z$ and $P_W$ are nonatomic (i.e., assign zero mass to singletons) and the learning algorithm is deterministic. To remove this drawback, we will use an elegant auxiliary randomization device introduced by Steinke and Zakynthinou [10].

Let $\varepsilon = (\varepsilon_1, \ldots, \varepsilon_n)$ be an $n$-tuple of i.i.d. Rademacher random variables, i.e., $\mathbf{P}[\varepsilon_i = \pm 1] = 1/2$, independent of $\tilde{S}$. For each $i$ let $\tilde{Z}_i^1 := Z_i$ and $\tilde{Z}_i^{-1} := Z'_i$ and let $\bar{P} = \bar{P}_{\tilde{S}\varepsilon W}$ be the joint probability law of $(\tilde{S}, \varepsilon, W)$, such that $\bar{P}_{\tilde{S}\varepsilon} = P_{\tilde{S}} \otimes P_\varepsilon$ and $\bar{P}_{W|\tilde{S}\varepsilon} := P_{W|\tilde{S}^\varepsilon}$ where $S^\varepsilon := (\tilde{Z}_1^{\varepsilon_1}, \ldots, \tilde{Z}_n^{\varepsilon_n})$. In other words, under $\bar{P}$, $\tilde{S}$ and $\varepsilon$ are independent and have their respective marginal distributions, while $W$ is generated by feeding the learning algorithm $P_{W|S}$ with the tuple $\tilde{S}^\varepsilon$. Consequently, $W$ is independent of $\tilde{S}^{-\varepsilon} = (\tilde{Z}_1^{-\varepsilon_1}, \ldots, \tilde{Z}_n^{-\varepsilon_n})$. Then, letting $P$ be the joint law of $(\tilde{S}, W)$, we have

$$\begin{aligned}
\mathbf{E}_P[|\text{gen}(W, S)|] &= \mathbf{E}_P \big| \mathbf{E}_P[L'_n(W) - L_n(W)|S, W] \big| \\
&\leq \mathbf{E}_P |L'_n(W) - L_n(W)| \\
&= \mathbf{E}_{\bar{P}} \Big| \frac{1}{n} \sum_{i=1}^n \left( \ell(W, \tilde{Z}_i^{-\varepsilon_i}) - \ell(W, \tilde{Z}_i^{\varepsilon_i}) \right) \Big| \\
&= \mathbf{E}_{\bar{P}} \Big| \frac{1}{n} \sum_{i=1}^n \varepsilon_i \left( \ell(W, Z'_i) - \ell(W, Z_i) \right) \Big|.
\end{aligned}$$

Thus, all the analysis can be carried out w.r.t. $\bar{P}$, as in the following:

**Theorem 2.** *Assume there exists a function $\Delta : \mathcal{Z} \times \mathcal{Z} \to \mathbb{R}_+$, such that $|\ell(w, z) - \ell(w, z')| \leq \Delta(z, z')$ for all $w \in \mathcal{W}$ and $z, z' \in \mathcal{Z}$. Then for any Markov kernel $Q_{W|\tilde{S}}$ with access to $\tilde{S}$ but not to $\varepsilon$ we have*

$$\mathbf{E}_P[|\text{gen}(W, S)|] \leq \frac{\sqrt{12}}{n} \mathbf{E}_{\bar{P}} \left[ \|\Delta(\tilde{S})\|_{\ell^2} \left( \psi_2^{-1}\left( \frac{\mathrm{d}\bar{P}_{W|\tilde{S}\varepsilon}}{\mathrm{d}Q_{W|\tilde{S}}} \right) + 1 \right) \right], \tag{8}$$

*where $\|\Delta(\tilde{s})\|_{\ell^2} := \left( \sum_{i=1}^n \Delta(z_i, z'_i)^2 \right)^{1/2}$.*

The same assumption on $\ell$ was also made in [10]. Optimizing over $Q_{W|\tilde{S}}$, we can recover their Theorem 5.1 (again, up to a multiplicative constant and a $O(1/\sqrt{n})$ fluctuation term):

**Corollary 2.** *Under the assumptions of Theorem 2,*

$$\mathbf{E}_P[|\mathrm{gen}(W,S)|] \leq \sqrt{\frac{24}{n}\mathbf{E}[\Delta^2(Z,Z')]\big(I(W;\varepsilon|\tilde{S})+4\big)}, \qquad (9)$$

*where $Z$ and $Z'$ are independent samples from $P_Z$ and where the conditional mutual information is computed w.r.t. $\bar{P}$.*

The main advantage of using conditional mutual information is that it never exceeds $n\log 2$ (of course, the bound is only useful if $I(W;\varepsilon|\tilde{S}) = o(n)$).

## 5 Estimates using couplings

We now turn to the analysis of $\mathbf{E}[\mathrm{gen}(W,S)]$ using couplings. The starting point is the following observation: With $(S',S,W)$ be constructed as before, consider the quantities

$$\tilde{L}_n(w) := L'_n(w) - L_n(w) \equiv \frac{1}{n}\sum_{i=1}^n \big(\ell(w,Z'_i) - \ell(w,Z_i)\big).$$

Then, using the fact that $\langle P_{\tilde{S}} \otimes Q_W, \tilde{L}_n \rangle = 0$ for any $Q_W \in \mathcal{P}(\mathcal{W})$, we have

$$\mathbf{E}[\mathrm{gen}(W,S)] = \langle P_{\tilde{S}} \otimes P_{W|S}, \tilde{L}_n \rangle - \langle P_{\tilde{S}} \otimes Q_W, \tilde{L}_n \rangle$$
$$= \int_{\mathcal{Z}\times\mathcal{Z}} P_{\tilde{S}}(\mathrm{d}\tilde{s})\big(\langle P_{W|S=s}, \tilde{L}_n \rangle - \langle Q_W, \tilde{L}_n \rangle\big). \qquad (10)$$

This suggests the idea of introducing, for each $s \in \mathcal{Z}^n$, a coupling of $P_{W|S=s}$ and $Q_W$, i.e., a probability law $P_{UV|S=s}$ for a random element $(U,V)$ of $\mathcal{W}\times\mathcal{W}$ with marginals $P_U = P_{W|S=s}$ and $P_V = Q_W$. We then have the following:

**Theorem 3.** *For $u,v \in \mathcal{W}$ and $\tilde{s} = (s,s') \in \mathcal{Z}^n \times \mathcal{Z}^n$, define*

$$\sigma^2(u,v,\tilde{s}) := \sum_{i=1}^n \Big(\big(\ell(u,z'_i) - \ell(v,z'_i)\big) - \big(\ell(u,z_i) - \ell(v,z_i)\big)\Big)^2. \qquad (11)$$

*Then, for any $Q_W \in \mathcal{P}(\mathcal{W})$, any family of couplings $P_{UV|S=s} \in \Pi(P_{W|S=s}, Q_W)$ depending measurably on $s \in \mathcal{Z}^n$, and any $\mu_{UV} \in \mathcal{P}(\mathcal{W}\times\mathcal{W})$,*

$$\mathbf{E}[\mathrm{gen}(W,S)] \leq \frac{\sqrt{24}}{n}\mathbf{E}\left[\sigma(U,V,\tilde{S})\psi_2^{-1}\left(\frac{\mathrm{d}P_{UV|S}}{\mathrm{d}\mu_{UV}}\right) + \sqrt{\mathbf{E}[\sigma^2(\bar{U},\bar{V},\tilde{S})|\tilde{S}]}\right], \qquad (12)$$

*where the expectation on the right-hand side is w.r.t. the joint law of $(U,V,\bar{U},\bar{V},\tilde{S})$, under which $(S,U,V)$ are distributed according to $P_S \otimes P_{UV|S}$, $(\bar{U},\bar{V})$ are distributed according to $\mu_{UV}$ independently of $(U,V,S)$, and $S'$ is distributed according to $P_S$ independently of everything else.*

The proof makes essential use of symmetrization using an auxiliary $n$-tuple $\varepsilon$ of i.i.d. Rademacher random variables, which allows us to apply Lemma 1 conditionally on $\tilde{S}$.

The coupling-based formulation looks rather complicated compared to the setting of Section 4. However, being able to choose not just the "prior" $Q_W$, but also the couplings $P_{UV|S}$ of $P_{W|S}$ and $Q_W$ and the reference measure $\mu_{UV}$, allows us to overcome some of the shortcomings of the set-up of Section 4. Consider, for example, the case when the learning algorithm ignores the data, i.e., $P_{W|S} = P_W$. Then we can choose $Q_W = P_W$, $P_{UV|S}(\mathrm{d}u,\mathrm{d}v) = P_W(\mathrm{d}u) \otimes \delta_u(\mathrm{d}v)$, where $\delta_u$ is the Dirac measure concentrated on the point $u$, and $\mu_{UV} = P_{UV}$ (since the latter does not depend on $S$). With these choices, $U = V$ and $\bar{U} = \bar{V}$ almost surely, so the right-hand side of (12) is identically zero. By contrast, the bounds of Theorems 1 and 2 always include an additional $O(1/\sqrt{n})$ term even when $W$ and $\tilde{S}$ are independent.

Moreover, Theorem 3 can be used to recover the bounds of Theorems 1 and 2 up to multiplicative constants. For example, to recover Theorem 1, we apply Theorem 3 with $P_{UV|S} = P_{W|S} \otimes Q_W$, $\mu_{UV} = Q_W \otimes Q_W$, and with an estimate on $\sigma(U, V, \tilde{S})$ based on the subgaussianity of $\ell(w, Z)$.

For a more manageable bound that will be useful later, let us define the following for $u, v \in \mathcal{W}$:

$$d_{S,\ell}(u,v) := \left( \frac{1}{n} \sum_{i=1}^{n} \left( \ell(u, Z_i) - \ell(v, Z_i) \right)^2 \right)^{1/2} \equiv \|\ell(u, \cdot) - \ell(v, \cdot)\|_{L^2(P_n)}$$

$$d_{\ell}(u,v) := \left( \mathbf{E}\left[ \left( \ell(u, Z) - \ell(v, Z) \right)^2 \right] \right)^{1/2} \equiv \|\ell(u, \cdot) - \ell(v, \cdot)\|_{L^2(P_Z)},$$

**Corollary 3.** *Under the assumptions of Theorem 3,*

$$\mathbf{E}[\mathrm{gen}(W, S)] \leq \sqrt{\frac{48}{n}} \mathbf{E}\left[ \left( d_{\ell}(U, V) + d_{S,\ell}(U, V) \right) \psi_2^{-1}\left( \frac{\mathrm{d}P_{UV|S}}{\mathrm{d}\mu_{UV}} \right) + d_{\ell}(\bar{U}, \bar{V}) \right].$$

## 6 Refined estimates via chaining in the space of measures

We now combine the use of couplings as in Section 5 with a chaining argument. The basic idea is as follows: Instead of coupling $P_{W|S}$ with $Q_W$ directly, we interpolate between them using a (possibly infinite) sequence of Markov kernels $P_{W|S}^0, P_{W|S}^1, \ldots, P_{W|S}^K$, such that $P_{W|S}^0 = Q_W$ and $P_{W|S}^K = P_{W|S}$ (or $\lim_{k \to \infty} P_{W|S}^k = P_{W|S}$ in an appropriate sense, e.g., weakly for each $S$, if the sequence is infinite). Given any such sequence, we telescope the terms in (10) as follows:

$$\mathbf{E}[\mathrm{gen}(W, S)] = \int_{\mathcal{Z} \times \mathcal{Z}} P_{\tilde{S}}(\mathrm{d}\tilde{s}) \sum_{k=1}^{K} \left( \langle P_{W|S=s}^k, \tilde{L}_n \rangle - \langle P_{W|S=s}^{k-1}, \tilde{L}_n \rangle \right).$$

For each $k$, we can now choose a family of random couplings $P_{W_k W_{k-1}|S} \in \Pi(P_{W|S}^k, P_{W|S}^{k-1})$ and a deterministic probability measure $\rho_{W_k W_{k-1}} \in \mathcal{P}(\mathcal{W} \times \mathcal{W})$. The following is an immediate consequence of applying Corollary 3 to each summand:

**Theorem 4.** *Let $P_{W|S}$, $Q_W$, $P_{W_k W_{k-1}|S}$, and $\rho_{W_k W_{k-1}}$ be given as above. Then*

$\mathbf{E}[\mathrm{gen}(W, S)]$
$$\leq \sqrt{\frac{48}{n}} \sum_{k=1}^{K} \mathbf{E}\left[ \left( d_{\ell}(W_k, W_{k-1}) + d_{S,\ell}(W_k, W_{k-1}) \right) \psi_2^{-1}\left( \frac{\mathrm{d}P_{W_k W_{k-1}|S}}{\mathrm{d}\rho_{W_k W_{k-1}}} \right) + d_{\ell}(\bar{W}_k, \bar{W}_{k-1}) \right],$$

*where in the kth term on the right-hand side $(S, W_k, W_{k-1})$ are jointly distributed according to $P_S \otimes P_{W_k W_{k-1}|S}$ and $(\bar{W}_k, \bar{W}_{k-1})$ are jointly distributed according to $\rho_{W_k W_{k-1}}$.*

Apart from Theorem 1, we have been imposing only minimal assumptions on $\ell$ and then using symmetrization to construct various subgaussian random variables conditionally on $W$ and $\tilde{S}$. For the next series of results, we will assume something more, namely that $(\mathcal{W}, d)$ is a metric space and that the following holds for the centered loss $\bar{\ell}(w, z) := \ell(w, z) - \mathbf{E}[\ell(w, Z)]$:

$$\left\| \sum_{i=1}^{n} (\bar{\ell}(u, Z_i) - \bar{\ell}(v, Z_i)) \right\|_{\psi_2} \leq \sqrt{n} d(u, v), \quad \forall u, v \in \mathcal{W}. \tag{13}$$

In other words, the centered empirical process $\frac{1}{\sqrt{n}} \sum_{i=1}^{n} \bar{\ell}(w, Z_i)$ indexed by the elements of $(\mathcal{W}, d)$ is a subgaussian process [1–3].

**Theorem 5.** *Assume (13). Then*

$$\mathbf{E}[\mathrm{gen}(W, S)] \leq \sqrt{\frac{2}{n}} \sum_{k=1}^{K} \mathbf{E}\left[ d(W_k, W_{k-1}) \psi_2^{-1}\left( \frac{\mathrm{d}P_{W_k W_{k-1}|S}}{\mathrm{d}\rho_{W_k W_{k-1}}} \right) + d(\bar{W}_k, \bar{W}_{k-1}) \right] \tag{14}$$

As a byproduct, we recover the stochastic chaining bounds of Zhou et al. [20] (which, in turn, subsume the bounds of Asadi et al. [18]):

**Corollary 4.** *Let $P_Z$ and $P_{W|S}$ be given, and let $P_W$ be the marginal law of $W$. Let $\left(P_{W_k|S}\right)_{k\geq 0}$ be a sequence of Markov kernels satisfying the following conditions: (i) $P_{W_0|S} = P_W$; (ii) $P_{W_k|S} \xrightarrow{k\to\infty} P_{W|S}$; (iii) for every $k \geq 1$, $S - W - W_k - W_{k-1}$ is a Markov chain. Then*

$$\mathbf{E}[\mathrm{gen}(W,S)] \leq \sqrt{\frac{2}{n}} \sum_{k=1}^{\infty} \mathbf{E}\Big[d(W_k, W_{k-1})\Big(\sqrt{D(P_{S|W_k}\|P_S)} + 1\Big)\Big] \tag{15}$$

$$\leq \sqrt{\frac{2}{n}} \sum_{k=1}^{\infty} \sqrt{\mathbf{E}[d^2(W_k, W_{k-1})]}\big(\sqrt{I(W_k; S)} + 2\big). \tag{16}$$

Finally, we give an estimate based on 2-Wasserstein distances (cf. Section 2 for definitions and notation). Let $\mathsf{W}_2(\cdot, \cdot)$ be the 2-Wasserstein distance on $\mathcal{P}_2(\mathcal{W})$ induced by the metric $d$ on $\mathcal{W}$. A (constant-speed) *geodesic* connecting two probability measures $P, Q \in \mathcal{P}_2(\mathcal{W})$ is a continuous path $[0,1] \ni t \mapsto \rho_t \in \mathcal{P}_2(\mathcal{W})$, such that $\rho_0 = P$, $\rho_1 = Q$, and $\mathsf{W}_2(\rho_s, \rho_t) = (t-s)\mathsf{W}_2(P, Q)$ for all $0 \leq s \leq t \leq 1$ [22,23]. Then we have the following corollary of Theorem 5:

**Corollary 5.** *Let $P_Z$ and $P_{W|S}$ be given, and let $P_W$ be the marginal law of $W$. With respect to 2-Wasserstein distance, let $(P_{W_k|S})_{0\leq k\leq K}$ be some points on the constant-speed geodesic $(\rho_t)_{t\in[0,1]}$ with endpoints $\rho_0 = P_{W_0|S} = P_{W|S}$ and $\rho_1 = P_{W_K|S} = P_W$ (where $K$ may be infinite), i.e., there exist some $t_0 = 0 < t_1 < \cdots < t_k < \cdots < t_K = 1$, such that $P_{W_k|S} = \rho_{t_k}$ for $k = 0, 1, \ldots$. For each $k$ let $P_{W_k W_{k-1}|S}$ be the optimal coupling between the neighboring points $P_{W_{k-1}|S}$ and $P_{W_k|S}$, i.e., the one that achieves $\mathsf{W}_2(P_{W_{k-1}|S}, P_{W_k|S})$. Then*

$$\mathbf{E}[\mathrm{gen}(W,S)]$$
$$\leq \sqrt{\frac{2}{n}}\left(2\,\mathbf{E}[\mathsf{W}_2(P_{W|S}, P_W)] + \sum_{k=1}^{K} \mathbf{E}\Big[\mathsf{W}_2(P_{W_k|S}, P_{W_{k-1}|S})\sqrt{D(P_{W_k W_{k-1}|S}\|P_{W_k W_{k-1}})}\Big]\right). \tag{17}$$

Observe that the first term on the right-hand side of (17) is the expected 2-Wasserstein distance between the posterior $P_{W|S}$ and the prior $P_W$, while the second term is a sum of "divergence weighted" Wasserstein distances. Also note that the form of the second term is in the spirit of the Dudley entropy integral [1–3], where the Wasserstein distance corresponds to the radius of the covering ball and the square root of the divergence corresponds to square root of the metric entropy. We should also point out that the result in Corollary 5 does not require Lipschitz continuity of the loss function $\ell(w, z)$ w.r.t. the hypothesis $w \in \mathcal{W}$, except in a weaker stochastic sense as in (13), in contrast to some existing works that obtain generalization bounds using Wasserstein distances [27,28].

## 7 Tail estimates

Next, we turn to high-probability tail estimates on $\mathrm{gen}(W, S)$. We start with the following simple observation: Assume $\ell(w, Z)$ is $\sigma$-subgaussian for all $w \in \mathcal{W}$ when $Z \sim P_Z$. Then, for any $Q_W \in \mathcal{P}(\mathcal{W})$ such that $P_{W|S=s} \ll Q_W$ for all $s \in \mathcal{Z}^n$, we have

$$\mathbf{E}\left[\exp\left(\frac{\mathrm{gen}^2(W,S)}{6\sigma^2/n} - \log\Big(1 + \frac{\mathrm{d}P_{W|S}}{\mathrm{d}Q_W}(W)\Big)\right)\right] \leq \mathbf{E}\left[\exp\left(\frac{\mathrm{gen}^2(\bar{W}, S)}{6\sigma^2/n}\right)\right] \leq 1$$

with $\bar{W} \sim Q_W$ independent of $(S, W)$. Thus, by Markov's inequality, for any $0 < \delta < 1$,

$$\mathbf{P}\left[|\mathrm{gen}(W,S)| > \sigma\sqrt{\frac{6}{n}}\left(\psi_2^{-1}\Big(\frac{\mathrm{d}P_{W|S}}{\mathrm{d}Q_W}(W)\Big) + \sqrt{\log\frac{1}{\delta}}\right)\right] \leq \delta.$$

In other words, $|\mathrm{gen}(W, S)| \lesssim \frac{\sigma}{\sqrt{n}}\psi_2^{-1}\big(\frac{\mathrm{d}P_{W|S}}{\mathrm{d}Q_W}\big)$ with high $P_{SW}$-probability. Similar observations are made by Hellström and Durisi [9] with $Q_W = P_W$, giving high-probability bounds of the form $|\mathrm{gen}(W, S)| \lesssim \sqrt{\frac{\sigma^2 D(P_{W|S}\|P_W)}{n}}$. Generalization bounds in terms of the divergence $D(P_{W|S}\|P_W)$ are also common in the PAC-Bayes literature [4,5]. Moreover, using the inequality (5) in Lemma 1, we can give high $P_S$-probability bounds on the conditional expectation

$$\langle P_{W|S}, |\mathrm{gen}(W,S)|\rangle = \langle P_{W|S}, |L(W) - L_n(W)|\rangle.$$

**Theorem 6.** *Assume $\ell(w, Z)$ is $\sigma$-subgaussian for all $w$ when $Z \sim P_Z$. Then, for any $Q_W \in \mathcal{P}(\mathcal{W})$, the following holds with $P_S$-probability of at least $1 - \delta$:*

$$\langle P_{W|S}, |\mathrm{gen}(W, S)| \rangle \leq \sqrt{\frac{24\sigma^2}{n} \left( \left\langle P_{W|S}, \psi_2^{-1}\left(\frac{\mathrm{d}P_{W|S}}{\mathrm{d}Q_W}\right) \right\rangle + 1 + \sqrt{\log\frac{2}{\delta}} \right)}.$$

Another type of result that appears frequently in the literature on PAC-Bayes methods pertains to so-called *transductive bounds*, i.e., inequalities for the difference between

$$\langle P_n' \otimes P_{W|S}, \ell \rangle - \langle P_n' \otimes Q_W, \ell \rangle \equiv \frac{1}{n} \sum_{i=1}^n \mathbf{E}[\ell(W, Z_i') - \ell(\bar{W}, Z_i')|\tilde{S}],$$

and

$$\langle P_n \otimes P_{W|S}, \ell \rangle - \langle P_n \otimes Q_W, \ell \rangle \equiv \frac{1}{n} \sum_{i=1}^n \mathbf{E}[\ell(W, Z_i) - \ell(\bar{W}, Z_i)|\tilde{S}],$$

where $Q_W$ is some fixed "prior" and where $\bar{W} \sim Q_W$ is independent of $(S', S, W)$. Using our techniques, we can give the following general transductive bound:

**Theorem 7.** *Let $P_{W|S}$ and $Q_W$ be given and take any $(P_{W_k W_{k-1}|S})_{k=1}^K$ and $(\rho_{W_k W_{k-1}})_{k=1}^K$ as in Theorem 4. Also, let $\mathbf{p} = (p_1, p_2, \dots)$ be a strictly positive probability distribution on $\mathbb{N}$. Then the following holds with $P_{\tilde{S}}$-probability at least $1 - \delta$:*

$$\left( \langle P_n' \otimes P_{W|S}, \ell \rangle - \langle P_n' \otimes Q_W, \ell \rangle \right) - \left( \langle P_n \otimes P_{W|S}, \ell \rangle - \langle P_n \otimes Q_W, \ell \rangle \right)$$

$$\leq \sqrt{\frac{96}{n}} \sum_{k=1}^K \left( \sqrt{\langle \rho_{W_k W_{k-1}}, d_{\tilde{S}, \ell}^2 \rangle} + \left\langle P_{W_k W_{k-1}|S}, d_{\tilde{S}, \ell} \psi_2^{-1}\left(\frac{\mathrm{d}P_{W_k W_{k-1}|S}}{\mathrm{d}\rho_{W_k W_{k-1}}}\right) \right\rangle \right.$$

$$\left. + \langle P_{W_k W_{k-1}|S}, d_{\tilde{S}, \ell} \rangle \sqrt{\log\frac{2}{p_k \delta}} \right),$$

*where*

$$d_{\tilde{S}, \ell}^2(u, v) := \frac{1}{2n} \sum_{i=1}^n \left( \left(\ell(u, Z_i) - \ell(v, Z_i)\right)^2 + \left(\ell(u, Z_i') - \ell(v, Z_i')\right)^2 \right).$$

This result subsumes some existing transductive PAC-Bayes estimates, such as Theorem 2 of Audibert and Bousquet [17]. Let us briefly explain how we can recover this result from Theorem 7. Assume that $\mathcal{W}$ is countable and let $(\mathcal{A}_k)$ be an increasing sequence of finite partitions of $\mathcal{W}$ with $\mathcal{A}_0 = \{\mathcal{W}\}$. For each $k$ and each $w \in \mathcal{W}$, let $A_k(w)$ be the unique set in $\mathcal{A}_k$ containing $w$. Choose a representative point in each $A \in \mathcal{A}_k$ and let $\mathcal{W}_k$ denote the set of all such representatives, with $\mathcal{W}_0 = \{w_0\}$. Take $P_{W_\infty|S} = P_{W|S}$ and $P_{W_0} = Q_W = \delta_{w_0}$. Now, for each $k \geq 0$, we take $P_{W_k|S}$ as the *projection* of $P_{W|S}$ onto $\mathcal{W}_k$, i.e., the finite mixture

$$P_{W_k|S} := \sum_{w \in \mathcal{W}_k} P_{W|S}(A_k(w)) \delta_w.$$

Moreover, given some "prior" $\pi \in \mathcal{P}(\mathcal{W})$, we can construct a sequence $(\pi_k)_{k=0}^\infty$ of probability measures with $\pi_\infty = \pi$ and $\pi_0 = \delta_{w_0}$, such that $\pi_k$ is a projection of $\pi$ onto $\mathcal{W}_k$. Now observe that, for each $k$, $S - W_\infty - W_k - W_{k-1}$ is a Markov chain. Indeed, if we know $P_{W_k|S}$, then we can construct $P_{W_\ell|S}$ for any $\ell < k$ without knowledge of $S$. With these ingredients in place, let us choose $P_{W_k W_{k-1}|S} = P_{W_{k-1}|W_k} \otimes P_{W_k|S}$ and $\rho_{W_k W_{k-1}} = \pi_k \otimes P_{W_{k-1}|W_k}$. Then, using Cauchy–Schwarz and Jensen, we conclude that the following holds with $P_{\tilde{S}}$-probability at least $1 - \delta$:

$$\left( \langle P_n' \otimes P_{W|S}, \ell \rangle - \langle P_n' \otimes \delta_{w_0}, \ell \rangle \right) - \left( \langle P_n \otimes P_{W|S}, \ell \rangle - \langle P_n \otimes \delta_{w_0}, \ell \rangle \right)$$

$$\leq \sqrt{\frac{96}{n}} \sum_{k=1}^\infty \left( \sqrt{\langle \pi_k \otimes P_{W_{k-1}|W_k}, d_{\tilde{S}, \ell}^2 \rangle} \right.$$

$$\left. + \sqrt{2 \langle P_{W_k|S} \otimes P_{W_{k-1}|W_k}, d_{\tilde{S}, \ell}^2 \rangle \left( D(P_{W_k|S} \| \pi_k) + \log\frac{2e}{p_k \delta} \right)} \right).$$

This recovers [17, Thm. 2] up to an extra term that scales like $\frac{1}{\sqrt{n}} \sum_k \sqrt{\langle \pi_k \otimes P_{W_{k-1}|W_k}, d_{\tilde{S}, \ell}^2 \rangle}$.

# 8 The Fernique–Talagrand bound

As a bonus, we show that a combination of decorrelation and chaining in the space of measures can be used to recover the upper bounds of Fernique [29] and Talagrand [30] on the expected supremum of a stochastic process in terms of majorizing measures (see Eq. (19) below and also [31, 32]).

For simplicity, let $(T, d)$ be a finite metric space with $\operatorname{diam}(T) = \sup\{d(u,v) : u, v \in T\} < \infty$. Let $B(t, r)$ denote the ball of radius $r \geq 0$ centered at $t \in T$, i.e., $B(t, r) := \{u \in T : d(u, t) \leq r\}$. Let $(X_t)_{t \in T}$ be a centered stochastic process defined on some probability space $(\Omega, \mathcal{A}, \mathbb{P})$ and satisfying

$$\mathbf{E}\left[\psi_p\left(\frac{|X_u - X_v|}{d(u, v)}\right)\right] \leq 1, \qquad \forall u, v \in T \tag{18}$$

for some $p \geq 1$. Then we can obtain the following result using chaining in the space of measures and decorrelation estimates:

**Theorem 8.** *Let $\tau$ be a random element of $T$, i.e., a measurable map $\tau : \Omega \to T$ with marginal probability law $\nu$. Then for any $\mu \in \mathcal{P}(T)$ we have*

$$\mathbf{E}[X_\tau] \lesssim \operatorname{diam}(T) + \int_T \int_0^{\operatorname{diam}(T)} \left(\log \frac{1}{\mu(B(t, \varepsilon))}\right)^{1/p} \mathrm{d}\varepsilon\, \nu(\mathrm{d}t).$$

Applying Theorem 8 to $\tau^*$ defined in (1) and then minimizing over $\mu$, we recover a Fernique–Talagrand type bound on the expected supremum of $X_t$:

$$\mathbf{E}\left[\sup_{t \in T} X_t\right] = \mathbf{E}[X_{\tau^*}] \lesssim \operatorname{diam}(T) + \inf_{\mu \in \mathcal{P}(T)} \sup_{t \in T} \int_0^{\operatorname{diam}(T)} \left(\log \frac{1}{\mu(B(t, \varepsilon))}\right)^{1/p} \mathrm{d}\varepsilon. \tag{19}$$

# 9 Conclusion and future work

In this paper, we have presented a unified framework for information-theoretic generalization bounds based on a combination of two key ideas (decorrelation and chaining in the space of measures). However, our method has certain limitations, which we plan to address in future work. For example, it would be desirable to cover the case of processes satisfying Bernstein-type (mixed $\psi_1$ and $\psi_2$) increment conditions. It would also be of interest to see whether there are any connections to the convex-analytic approach of Lugosi and Neu [33]. Finally, since our method seamlessly interpolates between Fernique–Talagrand type bounds and information-theoretic bounds, we plan to use it to further develop the ideas of Hodgkinson et al. [21], who were the first to combine these two complementary approaches to analyze the generalization capabilities of iterative learning algorithms.

## Acknowledgments

This work was supported by the Illinois Institute for Data Science and Dynamical Systems (iDS$^2$), an NSF HDR TRIPODS institute, under award CCF–1934986. The authors would like to thank Matus Telgarsky for some valuable suggestions and the anonymous reviewers for pointing out some relevant work that was not cited in the original submission.

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

## A  Some elementary facts

We first list some useful inequalities for $\psi_p$ and $\psi_p^{-1}$. Note that the estimates may not be the sharpest, but they suffice for our needs.

**Proposition A.2.** *For $p \geq 1$ and $x \geq 0$, let $\psi_p(x) = \exp(x^p) - 1$ and let $\psi_p^{-1}(x) = (\log(x+1))^{1/p}$ be its inverse. Then we have the following:*

  *(i)* $\psi_p^2(x/2^{1/p}) \leq \psi_p(x)$.

  *(ii)* $x\psi_p(x/4^{1/p}) \leq 2^{1/p}\psi_p(x/2^{1/p})$.

  *(iii) for $x \geq 0$ and $q \geq 1$, $\psi_p^{-1}(x^q) \leq q^{1/p}\psi_p^{-1}(x)$.*

  *(iv) For $x \geq 1$, $\psi_p^{-1}(x) \leq (\log(x))^{1/p} + 1$.*

*Proof.*

(i) For any $x \geq 0$,
$$\psi_p(x) = \exp(x^p) - 1 = (\exp(x^p/2) - 1)(\exp(x^p/2) + 1) \geq (\exp(x^p/2) - 1)^2$$
$$= \psi_p^2(x/2^{1/p}).$$

(ii) We only need to consider the case $x \geq 1$ since otherwise the inequality is obvious. Since $y \leq 2(\exp(y/4) + 1)$ for all $y \geq 1$, we have
$$x \leq 2^{1/p}(\exp(x^p/4) + 1)^{1/p} \leq 2^{1/p}(\exp(x^p/4p) + 1) \leq 2^{1/p}(\exp(x^p/4) + 1).$$
Then
$$\begin{aligned} x\psi_p(x/4^{1/p}) &= x(\exp(x^p/4) - 1)\\ &\leq 2^{1/p}(\exp(x^p/4) + 1)(\exp(x^p/4) - 1)\\ &= 2^{1/p}(\exp(x^p/2) - 1)\\ &= 2^{1/p}\psi_p(x/2^{1/p}).\end{aligned}$$

(iii) Since $x \geq 0$ and $q \geq 1$,
$$\psi_p^{-1}(x^q) = (\log(1 + x^q))^{1/p} \leq (\log(1 + x)^q)^{1/p} = q^{1/p}\psi_p^{-1}(x).$$

(iv) When $x \geq 1$,
$$\mathrm{e}x \geq x + 1 \implies \log x + 1 \geq \log(x + 1) \implies \log^{1/p}(x) + 1 \geq \psi_p^{-1}(x).$$

$\square$

The following simple result is for converting between sums and integrals:

**Proposition A.3.** *For any $r \geq 2$, $K \in \mathbb{N}$, and a continuous nonincreasing $f : (0, +\infty) \to (0, +\infty)$, we have*
$$\sum_{k=1}^{K} r^{-k} f(r^{-k}) \leq r \int_0^1 f(\varepsilon)\,\mathrm{d}\varepsilon \leq r^2 \sum_{k=0}^{\infty} r^{-k} f(r^{-k}) \tag{A.1}$$

*Proof.* Using the monotonicity of $f$, we have
$$\sum_{k=1}^{K} r^{-k} f(r^{-k}) \leq \sum_{k=1}^{K} r^{-k}(r-1)f(r^{-k}) \leq r \sum_{k=1}^{K} \int_{r^{-k-1}}^{r^{-k}} f(\varepsilon)\,\mathrm{d}\varepsilon$$
$$\leq r \int_0^1 f(\varepsilon)\mathrm{d}\varepsilon \leq r \sum_{k=0}^{\infty} \int_{r^{-k}}^{r^{-k+1}} f(\epsilon)\,\mathrm{d}\varepsilon \leq r^2 \sum_{k=0}^{\infty} r^{-k} f(r^{-k}).$$

$\square$

# B  Omitted proofs

## B.1  Proofs for Section 2

**Proof of Proposition 1.**  It follows from the inequality $x \log(x + 1) \leq x \log x + 1$, $x \geq 0$, that

$$\frac{\mathrm{d}\mu}{\mathrm{d}\nu} \log\left(\frac{\mathrm{d}\mu}{\mathrm{d}\nu} + 1\right) \leq \frac{\mathrm{d}\mu}{\mathrm{d}\nu} \log \frac{\mathrm{d}\mu}{\mathrm{d}\nu} + 1.$$

Using this and Jensen's inequality, we get

$$\begin{aligned}
\left\langle \mu, \psi_p^{-1}\left(\frac{\mathrm{d}\mu}{\mathrm{d}\nu}\right)\right\rangle &= \left\langle \mu, \left(\log\left(\frac{\mathrm{d}\mu}{\mathrm{d}\nu} + 1\right)\right)^{1/p}\right\rangle \\
&\leq \left(\left\langle \mu, \log\left(\frac{\mathrm{d}\mu}{\mathrm{d}\nu} + 1\right)\right\rangle\right)^{1/p} \\
&= \left(\left\langle \nu, \frac{\mathrm{d}\mu}{\mathrm{d}\nu} \log\left(\frac{\mathrm{d}\mu}{\mathrm{d}\nu} + 1\right)\right\rangle\right)^{1/p} \\
&\leq \left(\left\langle \nu, \frac{\mathrm{d}\mu}{\mathrm{d}\nu} \log \frac{\mathrm{d}\mu}{\mathrm{d}\nu}\right\rangle + 1\right)^{1/p} \\
&= \left(D(\mu\|\nu) + 1\right)^{1/p}.
\end{aligned}$$

## B.2  Proofs for Section 3

**Proof of Lemma 1.**  To prove (4), we start with the Young-type inequality

$$xy \leq \psi_p^*(x) + \psi_p(y), \qquad x, y \geq 0$$

where

$$\psi_p^*(x) := \sup_{y \geq 0}\left(xy - \psi_p(y)\right)$$

is the (one-sided) Legendre–Fenchel conjugate of $\psi_p$. While a closed-form expression for $\psi_p^*$ is not available, we claim that we can bound it from above as $\psi_p^*(x) \leq 2^{1/p} x \psi_p^{-1}(x)$, resulting in

$$xy \leq 2^{1/p} x \psi_p^{-1}(x) + \psi_p(y). \tag{B.1}$$

To establish the claim, we write

$$\sup_{y \geq 0}\left(xy - \psi_p(y)\right) = \sup_{y \geq 0}\left(xy - (e^{y^p/2} - 1)(e^{y^p/2} + 1)\right)$$

and consider two cases:

- if $y \leq 2^{1/p}\psi_p^{-1}(x)$, then

$$xy - (e^{y^p/2} - 1)(e^{y^p/2} + 1) \leq 2^{1/p} x \psi_p^{-1}(x).$$

- if $y > 2^{1/p}\psi_p^{-1}(x)$, then

$$xy - (e^{y^p/2} - 1)(e^{y^p/2} + 1) \leq (e^{y^p/2} - 1)(y - (e^{y^p/2} + 1)) \leq 0.$$

Applying (B.1) with $x = \frac{\mathrm{d}\mu}{\mathrm{d}\nu}$ and $y = g$ gives

$$g\frac{\mathrm{d}\mu}{\mathrm{d}\nu} \leq 2^{1/p}\frac{\mathrm{d}\mu}{\mathrm{d}\nu}\psi_p^{-1}\left(\frac{\mathrm{d}\mu}{\mathrm{d}\nu}\right) + \psi_p(g),$$

so that

$$\begin{aligned}
\langle \mu, fg \rangle &= \left\langle \nu, fg\frac{\mathrm{d}\mu}{\mathrm{d}\nu}\right\rangle \\
&\leq \left\langle \nu, \left(2^{1/p}f\frac{\mathrm{d}\mu}{\mathrm{d}\nu}\psi_p^{-1}\left(\frac{\mathrm{d}\mu}{\mathrm{d}\nu}\right) + f\psi_p(g)\right)\right\rangle \\
&= 2^{1/p}\left\langle \mu, f\psi_p^{-1}\left(\frac{\mathrm{d}\mu}{\mathrm{d}\nu}\right)\right\rangle + \langle \nu, f\psi_p(g)\rangle.
\end{aligned}$$

To prove (5), define the event

$$E := \left\{ \frac{\mathrm{d}\mu}{\mathrm{d}\nu} \geq \frac{\exp(g^p/4) - 1}{\langle \nu, \exp(g^p) \rangle} \right\}.$$

Then, since $\langle \nu, \exp(g^p) \rangle \geq 1$,

$$\int_E fg \,\mathrm{d}\mu \leq 4^{1/p} \int f \left( \log \left( \frac{\mathrm{d}\mu}{\mathrm{d}\nu} \langle \nu, \exp(g^p) \rangle + 1 \right) \right)^{1/p} \mathrm{d}\mu$$

$$\leq 4^{1/p} \int f \left( \log \left( \frac{\mathrm{d}\mu}{\mathrm{d}\nu} + 1 \right) + \log\langle \nu, \exp(g^p) \rangle \right)^{1/p} \mathrm{d}\mu$$

$$\leq 4^{1/p} \int f \left( \log \left( \frac{\mathrm{d}\mu}{\mathrm{d}\nu} + 1 \right) \right)^{1/p} \mathrm{d}\mu + 4^{1/p} \int f \,\mathrm{d}\mu \cdot \left( \log\langle \nu, \exp(g^p) \rangle \right)^{1/p}$$

$$= 4^{1/p} \left\langle \mu, f\psi_p^{-1} \left( \frac{\mathrm{d}\mu}{\mathrm{d}\nu} \right) \right\rangle + 4^{1/p} \|f\|_{L^1(\mu)} \left( \log\langle \nu, \exp(g^p) \rangle \right)^{1/p}.$$

On the other hand,

$$\int_{E^c} fg \,\mathrm{d}\mu \leq \int fg \frac{\exp(g^p/4) - 1}{\langle \nu, \exp(g^p) \rangle} \,\mathrm{d}\nu$$

$$\leq 2^{1/p} \int f \frac{\exp(g^p/2)}{\langle \nu, \exp(g^p) \rangle} \,\mathrm{d}\nu$$

$$\leq 2^{1/p} \|f\|_{L^2(\nu)},$$

where the first inequality is by the definition of $E$, the second inequality follows from Proposition A.2(ii), and the third inequality is by Cauchy–Schwarz. Putting everything together, we get (5).

### B.3 Proofs for Section 4

**Proof of Theorem 1.** It follows from the independence of $Z_1, \ldots, Z_n$ that $\mathrm{gen}(w, S)$ is $(\sigma/\sqrt{n})$-subgaussian, so

$$\mathbf{E}\left[ \psi_2 \left( \frac{|\mathrm{gen}(w, S)|}{\sigma\sqrt{6/n}} \right) \right] \leq 1, \qquad \forall w \in \mathcal{W}. \tag{B.2}$$

Using Lemma 1 with $\mu = P_{W|S}$, $\nu = Q_W$, $f(w) = \sigma\sqrt{6/n}$, and $g(w) = \frac{|\mathrm{gen}(w, S)|}{\sigma\sqrt{6/n}}$, we have

$$\langle P_{W|S}, |\mathrm{gen}(\cdot, S)| \rangle \leq \sqrt{\frac{12\sigma^2}{n}} \left( \left\langle P_{W|S}, \psi_2^{-1} \left( \frac{\mathrm{d}P_{W|S}}{\mathrm{d}Q_W} \right) \right\rangle + \left\langle Q_W, \psi_2 \left( \frac{|\mathrm{gen}(\cdot, S|}{\sigma\sqrt{6/n}} \right) \right\rangle \right).$$

Taking expectations of both sides w.r.t. $P_S$ and using Fubini's theorem and (B.2), we get (6).

**Proof of Corollary 1.** Applying Proposition 1 conditionally on $S$ gives

$$\left\langle P_{W|S}, \psi_2^{-1} \left( \frac{\mathrm{d}P_{W|S}}{\mathrm{d}Q_W} \right) \right\rangle \leq \sqrt{D(P_{W|S}\|Q_W) + 1},$$

where the divergence $D(P_{W|S}\|Q_W)$, being a function of $S$, is a random variable. Substituting this into (6) and using Jensen's inequality, the definition of conditional divergence, and $\sqrt{a + b} \leq \sqrt{a} + \sqrt{b} \leq \sqrt{2(a + b)}$, we get

$$\mathbf{E}[|\mathrm{gen}(W, S)|] \leq \sqrt{\frac{24\sigma^2}{n} \left( D(P_{W|S}\|Q_W|P_S) + 4 \right)}.$$

Taking the infimum of both sides w.r.t. $Q_W$ and using (2), we get (7).

**Proof of Theorem 2.** For each fixed $(w, \tilde{s})$, the random variable $\delta(w, \tilde{s}, \varepsilon) := |\sum_{i=1}^n \varepsilon_i(\ell(w, z_i') - \ell(w, z_i))|$ is $\sigma(w, \tilde{s})$-subgaussian, where $\sigma(w, \tilde{s}) := \big(\sum_{i=1}^n (\ell(w, z_i') - \ell(w, z_i))^2\big)^{1/2}$. Thus,

$$\mathbf{E}_\varepsilon[\zeta(w, \tilde{s}, \varepsilon)] := \mathbf{E}_\varepsilon\left[\psi_2\left(\frac{\delta(w, \tilde{s}, \varepsilon)}{\sqrt{6}\sigma(w, \tilde{s})}\right)\right] \leq 1, \qquad \forall(w, \tilde{s}). \tag{B.3}$$

Applying Lemma 1 conditionally on $(\tilde{S}, \varepsilon)$ with $\mu = \bar{P}_{W|\tilde{S}\varepsilon}$, $\nu = Q_{W|\tilde{S}}$, $f(w) = \sigma(w, \tilde{S})$, $g(w) = \zeta(w, \tilde{S}, \varepsilon)$, we obtain

$$\langle \bar{P}_{W|\tilde{S}\varepsilon}, \sigma(\cdot, \tilde{S})\zeta(\cdot, \tilde{S}, \varepsilon)\rangle$$

$$\leq \sqrt{2}\left\langle \bar{P}_{W|\tilde{S}\varepsilon}, \sigma(\cdot, \tilde{S})\psi_2^{-1}\left(\frac{\mathrm{d}\bar{P}_{W|\tilde{S}, \varepsilon}}{\mathrm{d}Q_{W|\tilde{S}}}\right)\right\rangle + \left\langle Q_{W|\tilde{S}}, \sigma(\cdot, \tilde{S})\psi_2(\zeta(\cdot, \tilde{S}, \varepsilon))\right\rangle$$

$$\leq \sqrt{2}\|\Delta(\tilde{S})\|_{\ell^2}\left(\left\langle \bar{P}_{W|\tilde{S}\varepsilon}, \psi_2^{-1}\left(\frac{\mathrm{d}\bar{P}_{W|\tilde{S}, \varepsilon}}{\mathrm{d}Q_{W|\tilde{S}}}\right)\right\rangle + \left\langle Q_{W|\tilde{S}}, \psi_2(\zeta(\cdot, \tilde{S}, \varepsilon))\right\rangle\right).$$

Taking expectations of both sides w.r.t. $\tilde{S}$ and $\varepsilon$, then using Fubini's theorem, (B.3), and the inequality $\mathbf{E}_P[|\mathrm{gen}(W, S)|] \leq \frac{1}{n}\mathbf{E}_{\bar{P}}[\delta(W, \tilde{S}, \varepsilon)]$, we obtain (8).

**Proof of Corollary 2.** For any $Q_{W|\tilde{S}}$, using Proposition 1, Cauchy–Schwarz, and the independence of $(Z_i', Z_i)$, we have

$$\mathbf{E}_{\bar{P}}\left[\|\Delta(\tilde{S})\|_{\ell^2}\psi_2^{-1}\left(\frac{\mathrm{d}\bar{P}_{W|\tilde{S}, \varepsilon}}{\mathrm{d}Q_{W|\tilde{S}}}\right)\right]$$

$$\leq \sqrt{\mathbf{E}_{\bar{P}}[\|\Delta(\tilde{S})\|_{\ell^2}^2]\big(D(\bar{P}_{W|\tilde{S}\varepsilon}\|Q_{W|\tilde{S}}|\bar{P}_{\tilde{S}\varepsilon}) + 1\big)}$$

$$= \sqrt{n\mathbf{E}[\Delta(Z, Z')^2]\big(D(\bar{P}_{W|\tilde{S}\varepsilon}\|Q_{W|\tilde{S}}|\bar{P}_{\tilde{S}\varepsilon}) + 1\big)}.$$

Substituting this estimate into (8), taking the infimum of both sides w.r.t. $Q_{W|\tilde{S}}$, and using (3), we get (9).

## B.4 Proofs for Section 5

**Proof of Theorem 3.** Let

$$\delta(u, v, z, z') := \big(\ell(u, z') - \ell(v, z')\big) - \big(\ell(u, z) - \ell(v, z)\big),$$

$$\delta(u, v, \tilde{s}) := \sum_{i=1}^n \delta(u, v, z_i, z_i'),$$

$$\zeta(u, v, \tilde{s}) := \frac{|\delta(u, v, \tilde{s})|}{\sqrt{6}\sigma(u, v, \tilde{s})}.$$

For each fixed $(u, v) \in \mathcal{W}^2$, $\delta(u, v, Z_i, Z_i')$, $1 \leq i \leq n$, are i.i.d. symmetric random variables. Therefore, introducing a tuple $\varepsilon = (\varepsilon_1, \ldots, \varepsilon_n)$ of i.i.d. Rademacher random variables independent of everything else and using the fact that the joint distributions of $\big(\delta(u, v, Z_i, Z_i')\big)_{i=1}^n$ and $\big(\varepsilon_i\delta(u, v, Z_i, Z_i')\big)_{i=1}^n$ are the same, we see that

$$\mathbf{E}[\psi_2(\zeta(u, v, \tilde{S}))] = \mathbf{E}_{\tilde{S}}\mathbf{E}_\varepsilon\left[\psi_2\left(\frac{|\sum_{i=1}^n \varepsilon_i\delta(u, v, Z_i, Z_i')|}{\sqrt{6}\sigma(u, v, \tilde{S})}\right)\right] \leq 1,$$

where the inequality follows from the fact that, conditionally on $S$ and $S'$, the random variables $\sum_{i=1}^n \varepsilon_i\delta(u, v, Z_i, Z_i')$ are $\sigma(u, v, \tilde{S})$-subgaussian.

Now, given $Q_W \in \mathcal{P}(\mathcal{W})$ and a family of couplings $P_{UV|S=s} \in \Pi(P_{W|S=s}, Q_W)$, it follows from the above definitions and from (10) that

$$\mathbf{E}[\mathrm{gen}(W, S)] \leq \frac{1}{n}\mathbf{E}[|\delta(U, V, \tilde{S})|] = \frac{\sqrt{6}}{n}\mathbf{E}[\sigma(U, V, \tilde{S})\zeta(U, V, \tilde{S})]. \tag{B.4}$$

Picking any $\rho_{UV} \in \mathcal{P}(\mathcal{W} \times \mathcal{W})$ such that $P_{UV|S=s} \ll \rho_{UV}$ for all $s \in \mathcal{Z}^n$ and applying Lemma 1, we get

$$\langle P_{UV|S}, \sigma(\cdot, \tilde{S})\zeta(\cdot, \tilde{S}) \rangle$$
$$\leq 2 \left\langle P_{UV|S}, \sigma(\cdot, \tilde{S})\psi_2^{-1}\left(\frac{dP_{UV|S}}{d\rho_{UV}}\right) \right\rangle + \sqrt{2}\left\langle \rho_{UV}, \sigma(\cdot, \tilde{S})\psi_2\left(\frac{\zeta(\cdot, \tilde{S})}{\sqrt{2}}\right) \right\rangle.$$

Using the inequality $\psi_2^2(x/\sqrt{2}) \leq \psi_2(x)$ (see Proposition A.2(i)), Cauchy–Schwarz, and (B.4), we have

$$\mathbf{E}\left[\sigma(u, v, \tilde{S})\psi_2\left(\frac{\zeta(u, v, \tilde{S})}{\sqrt{2}}\right)\right] \leq \sqrt{\mathbf{E}[\sigma^2(u, v, \tilde{S})]}, \qquad \forall (u, v) \in \mathcal{W} \times \mathcal{W}.$$

Putting everything together and taking expectations w.r.t. $S$ and $S'$, we obtain (12).

**Proof of Corollary 3.** For $\sigma$ defined in Theorem 3, we have

$$\sigma^2(u, v, \tilde{S}) \leq 2\sum_{i=1}^{n}\left(\left(\ell(u, Z_i') - \ell(v, Z_i')\right)^2 + \left(\ell(u, Z_i) - \ell(v, Z_i)\right)^2\right).$$

Taking conditional expectations given $U, V, S$ and using Jensen's inequality gives

$$\mathbf{E}[\sigma(U, V, \tilde{S})|U, V, S] \leq \sqrt{\mathbf{E}[\sigma^2(U, V, \tilde{S})|U, V, S]}$$
$$\leq \sqrt{2n}\left(d_\ell(U, V) + d_{S,\ell}(U, V)\right).$$

An analogous argument gives

$$\sqrt{\mathbf{E}[\sigma^2(\bar{U}, \bar{V}, \tilde{S})|\bar{U}, \bar{V}]} \leq 2\sqrt{n}d_\ell(\bar{U}, \bar{V}).$$

Substituting these estimates into (12) gives the desired result.

### B.5 Proofs for Section 6

**Proof of Theorem 5.** Using the definition of $\bar{\ell}$, we have

$$\mathbf{E}[\text{gen}(W, S)] = \frac{1}{n}\sum_{k=1}^{K}\mathbf{E}\left[\sum_{i=1}^{n}\left(\bar{\ell}(W_k, Z_i) - \bar{\ell}(W_{k-1}, Z_i)\right)\right].$$

Applyng Lemma 1 conditionally on $S$ with $f(u, v) = d(u, v)$, $g(u, v) = \frac{|\sum_{i=1}^{n}(\bar{\ell}(u,,Z_i) - \bar{\ell}(v, Z_i))|}{\sqrt{n}d(u, v)}$, $\mu = P_{W_k W_{k-1}|S}$ and $\nu = \rho_{W_k W_{k-1}}$, taking expectations w.r.t. $P_S$, and using (13) gives the desired result.

**Proof of Corollary 4.** For each $k \geq 1$, let $\rho_{W_k W_{k-1}} = P_{W_k W_{k-1}}$. Then

$$\frac{dP_{W_k W_{k-1}|S}}{dP_{W_k W_{k-1}}} = \frac{dP_{W_k W_{k-1}S}}{d(P_{W_k W_{k-1}} \otimes P_S)} = \frac{dP_{S|W_k W_{k-1}}}{dP_S} = \frac{dP_{S|W_k}}{dP_S},$$

where we have made use of Bayes' rule and the fact that $S \perp\!\!\!\perp W_{k-1}|W_k$. Using this in (14) together with Proposition 1 gives (15). An application of Cauchy–Schwarz and Jensen gives (16).

**Proof of Corollary 5.** For each $k \geq 1$, let $\rho_{W_k W_{k-1}} = P_{W_k W_{k-1}}$. Notice that, by disintegration and the choice of couplings,

$$\mathbf{E}[d(\bar{W}_k, \bar{W}_{k-1})] = \mathbf{E}\left[\int d(u, v)P_{W_k W_{k-1}|S}(du, dv)\right] \leq \mathbf{E}[\mathsf{W}_2(P_{W_k|S}, P_{W_{k-1}|S})],$$

where we have used the fact that $\mathsf{W}_2(\cdot, \cdot)$ dominates $\mathsf{W}_1(\cdot, \cdot)$ [23]. Since $P_{W_k|S}$ are points on the geodesic connecting $P_{W|S}$ and $P_W$, we have

$$\sum_{k=1}^{K}\mathsf{W}_2(P_{W_k|S}, P_{W_{k-1}|S}) = \sum_{k=1}^{K}(t_k - t_{k-1})\mathsf{W}_2(P_{W|S}, P_W) = \mathsf{W}_2(P_{W|S}, P_W).$$

Using this together with Cauchy-Schwarz and Proposition 1, we obtain (17).

## B.6 Proofs for Section 7

**Proof of Theorem 6.** Applying (5) conditionally on $S$ with $f(w) = \sigma\sqrt{6/n}$, $g(w) = \frac{|\mathrm{gen}(w,S)|}{\sigma\sqrt{6/n}}$, $\mu = P_{W|S}$, and $\nu = Q_W$, we have

$$\langle P_{W|S}, |\mathrm{gen}(W,S)| \rangle$$
$$\leq \sqrt{\frac{24\sigma^2}{n}\left(1 + \left\langle P_{W|S}, \psi_2^{-1}\left(\frac{\mathrm{d}P_{W|S}}{\mathrm{d}Q_W}\right)\right\rangle + \left(\log\left\langle Q_W, \exp\left(\frac{\mathrm{gen}^2(\cdot,S)}{6\sigma^2/n}\right)\right\rangle\right)^{1/2}\right)}.$$

Since $\ell(w,S)$ is $(\sigma/\sqrt{n})$-subgaussian for all $w$, Markov's inequality gives, for any $0 < \delta < 1$,

$$\mathbf{P}\left[\left\langle Q_W, \exp\left(\frac{\mathrm{gen}^2(\cdot,S)}{6\sigma^2/n}\right)\right\rangle > \frac{2}{\delta}\right] \leq \frac{\delta}{2}\left\langle P_S \otimes Q_W, \exp\left(\frac{\mathrm{gen}^2(\cdot,\cdot)}{6\sigma^2/n}\right)\right\rangle \leq \delta,$$

which concludes the proof.

**Proof of Theorem 7.** The argument is almost identical to the proof of Theorem 3, with the difference that (5) is used for decorrelation.

To lighten the notation, let $\pi_k^S := P_{W_k W_{k-1}|S}$ and $\rho_k := \rho_{W_k W_{k-1}}$. Use the same definitions of $\delta, \sigma, \zeta$ as in the proof of Theorem 3. Then, applying (5) with $f(\cdot) = \sigma(\cdot, \tilde{S})$, $g(\cdot) = \zeta(\cdot, \tilde{S})$, $\mu = \pi_k^S$, $\nu = \rho_k$, we have

$$\langle \pi_k^S, \sigma(\cdot, \tilde{S})\zeta(\cdot, \tilde{S})\rangle \leq \sqrt{2}\|\sigma(\cdot, \tilde{S})\|_{L^2(\rho_k)} + 2\left\langle \pi_k^S, \sigma(\cdot, \tilde{S})\psi_2^{-1}\left(\frac{\mathrm{d}\pi_k^S}{\mathrm{d}\rho_k}\right)\right\rangle$$
$$+ 2\|\sigma(\cdot, \tilde{S})\|_{L^1(\pi_k^S)}\sqrt{\log\langle \rho_k, \exp\left(\zeta^2(\cdot, \tilde{S})\right)\rangle}.$$

By Markov's inequality and the union bound, for any $0 < \delta < 1$,

$$\mathbf{P}\left[\exists k \text{ s.t. } \left\langle \rho_k, \exp\left(\zeta^2(\cdot, \tilde{S})\right)\right\rangle > \frac{2}{p_k \delta}\right] \leq \sum_k \frac{p_k \delta}{2}\left\langle P_{\tilde{S}} \otimes \rho_k, \exp\left(\zeta^2(\cdot, \cdot)\right)\right\rangle \leq \delta.$$

Using this together with the estimate $\sigma(\cdot, \tilde{S}) \leq 2\sqrt{n}d_{\tilde{S},\ell}(\cdot)$ yields the result in the statement.

## B.7 Proofs for Section 8

**Proof of Theorem 8.** Without loss of generality, we assume $\mathrm{diam}(T) = 1$. Let $Q$ be the Markov kernel from $\Omega$ to $T$ defined by $Q(\cdot|\omega) = \delta_{\tau(\omega)}(\cdot)$; in particular, $\nu(\cdot) = \int_\Omega \mathbb{P}(\mathrm{d}\omega)Q(\cdot|\omega)$.

Fix some $r \geq 2$. For each $k \geq 0$ and each $t \in T$, let $B_k(t) := B(t, r^{-k})$. Since $T$ is finite, there exists some $K \in \mathbb{N}$, such that $B_K(t) = \{t\}$ for all $t \in T$. Let $\mu \in \mathcal{P}(T)$ be given. Define the following sequence of Markov kernels from $\Omega$ to $T$:

$$Q_k(\cdot|\omega) := \frac{\mu(\cdot \cap B_k(\tau(\omega)))}{\mu(B_k(\tau(\omega)))}, \qquad k = 0, \ldots, K.$$

Observe that $Q_0 = \mu$ and $Q_K = Q$. Then, since

$$\langle \mathbb{P} \otimes \mu, X\rangle = \int_{\Omega \times T} \mathbb{P}(\mathrm{d}\omega)\mu(\mathrm{d}t)X_t(\omega) = \int_T \mu(\mathrm{d}t)\mathbf{E}[X_t] = 0,$$

we can write

$$\mathbf{E}[X_\tau] = \langle \mathbb{P} \otimes Q, X\rangle - \langle \mathbb{P} \otimes \mu, X\rangle$$
$$= \langle \mathbb{P} \otimes Q_K, X\rangle - \langle \mathbb{P} \otimes Q_0, X\rangle$$
$$= \sum_{k=1}^K \langle \mathbb{P} \otimes Q_k - \mathbb{P} \otimes Q_{k-1}, X\rangle$$
$$= \sum_{k=1}^K \int_\Omega \left(\int_T X_t(\omega)Q_k(\mathrm{d}t|\omega) - \int_T X_t(\omega)Q_{k-1}(\mathrm{d}t|\omega)\right)\mathbb{P}(\mathrm{d}\omega)$$
$$\leq \sum_{k=1}^K \int_\Omega \int_{T \times T} |X_u(\omega) - X_v(\omega)|Q_k(\mathrm{d}u|\omega)Q_{k-1}(\mathrm{d}v|\omega)\mathbb{P}(\mathrm{d}\omega).$$

Applying (4) conditionally on $\omega$ with

$$f(u,v) = \mathbf{1}_{B_k(\tau(\omega))}(u)\mathbf{1}_{B_{k-1}(\tau(\omega))}(v)d(u,v),$$

$$g(u,v) = \frac{|X_u(\omega) - X_v(\omega)|}{d(u,v)},$$

$$\mu(\mathrm{d}u, \mathrm{d}v) = Q_k(\mathrm{d}u|\omega) \otimes Q_{k-1}(\mathrm{d}v|\omega),$$

$$\nu(\mathrm{d}u, \mathrm{d}v) = \mu(\mathrm{d}u) \otimes \mu(\mathrm{d}v),$$

and using the fact that $Q_k(\cdot|\omega)$ is supported on $B_k(\tau(\omega))$ and $B_k(\tau(\omega)) \subseteq B_{k-1}(\tau(\omega))$, we have

$$\int_{T \times T} |X_u(\omega) - X_v(\omega)|Q_k(\mathrm{d}u|\omega)Q_{k-1}(\mathrm{d}v|\omega)$$

$$\leq 2^{1/p}r^{-k+1}\psi_p^{-1}\left(\frac{1}{\mu(B_k(\tau(\omega)))^2}\right) + r^{-k+1}\int_{T \times T} \psi_p\left(\frac{|X_u(\omega) - X_v(\omega)|}{d(u,v)}\right)\mu(\mathrm{d}u)\mu(\mathrm{d}v)$$

$$\leq 2^{2/p}r^{-k+1}\psi_p^{-1}\left(\frac{1}{\mu(B_k(\tau(\omega)))}\right) + r^{-k+1}\int_{T \times T} \psi_p\left(\frac{|X_u(\omega) - X_v(\omega)|}{d(u,v)}\right)\mu(\mathrm{d}u)\mu(\mathrm{d}v),$$

where the first term in the last step is due to Proposition A.2(iii). Then, using the increment condition and Proposition A.3, we have

$$\mathbf{E}[X_\tau] \leq 2^{2/p}\sum_{k=1}^{K} r^{-k+1}\int_T \psi_p^{-1}\left(\frac{1}{\mu(B(t, r^{-k}))}\right)\nu(\mathrm{d}t) + \sum_{k=1}^{K} r^{-k+1}$$

$$\leq 1 + 2^{2/p}r^2\int_T\int_0^1 \psi_p^{-1}\left(\frac{1}{\mu(B(t,\varepsilon))}\right)\mathrm{d}\varepsilon\,\nu(\mathrm{d}t).$$

Since $1/\mu(B(t,\varepsilon)) \geq 1$, we can apply Proposition A.2(iv) to obtain the inequality

$$\mathbf{E}[X_\tau] \leq 2^{2/p}r^2\left(2 + \int_T\int_0^1\left(\log\frac{1}{\mu(B(t,\varepsilon))}\right)^{1/p}\mathrm{d}\varepsilon\,\nu(\mathrm{d}t)\right).$$

We can now take $r = 2$ to get the desired result when $\mathrm{diam}(T) = 1$; the general finite-diameter case follows by straightforward rescaling.

