# OpenReview forum: "A unified framework for information-theoretic generalization bounds"
_NeurIPS.cc/2023/Conference — NeurIPS 2023 poster_

### Official Review · Reviewer_HuV6 · 2023-06-26

**Soundness:** 3 good
**Presentation:** 3 good
**Contribution:** 2 fair
**Rating:** 6
**Confidence:** 3

**Summary:**

The paper studies the generalization error of the statistical learning algorithms from the information-theoretic point of view. In particular, by leveraging a “decorrelation lemma”, the authors show how various previously established upper bounds on the generalization error, including the common information-theoretic bounds, PAC-Bayes bounds, and chaining versions of them, can be recovered using this decorrelation lemma; and therefore, present a unifying methodology for establishing these bounds.

**Strengths:**

The authors using the introduced decorrelation lemma managed to compactly recover various existing upper bounds on the generalization error. In particular, the method naturally can be combined with the chaining methods to recover a number of results such as the generalization bounds of Zhou et al., Asadi et al., and the bound of Fernique and Talagrand on the expected supremum of a stochastic process. Moreover, the work presents high probability bounds; which is rarely common in the information-theoretic bounds on the generalization.

I believe the decorrelation lemma is a powerful technical lemma using which hopefully, new understandings and bounds can be achieved in future works. For this reason, I think the paper could be of interest to the part of the Neurips audiences that are interested in the theoretical understating of statistical learning algorithms.


**Weaknesses:**

1. The main weakness of the work is that it remains only at the technical level.

-  1.a. The form of the general bounds, e.g. Theorem 3, does not seem to be easy to compute. It is not clear whether the general results can be leveraged to derive new bounds or to give a new understanding of the generalization error of the learning algorithms (in general or for a particular algorithm). Can authors provide cases where the results of this work give “new” and “simple” or "practical" results? I think the authors would agree that the justification of how coupling in Theorem 3 can be used to show that the expected generalization error is zero when $P_{W|S}=P_W$ is not convincing enough to show the utility of that.

- 1.b. The only cases where these general results give concrete results are when they recover previous results (e.g. Information-theoretic bounds or PAC-Bayes ones) but only with the worst coefficients.

- 1.c. Moreover, even at the intuition level, it is not clear if this work, and more specifically, this decorrelation lemma, gives any new “understanding" of the generalizability of the learning algorithms. Does this work, and this decorrelation lemma, can provide any new intuition/understanding?
For these reasons, unfortunately, the paper remains only on the “technical level” and we can only hope that this interesting tool can be used for establishing new concrete results/understandings.

2. The paper lacks a proper literature review and comparison with other works. Indeed, only 25 works have been cited; most of which are purely listed in the very short introduction. The authors are needed to widen the comparison of the current work with previous literature on generalization error (including different information-theoretic approaches, PAC-Bayes ones, compression-based ones, etc.) to make a clear picture of where this work/tools stand.

3. As an example of the previous point, perhaps, as claimed by the authors, one of the main advantages of this work is the unifying methodology for deriving various existing bounds. This is indeed interesting, as has been done in some other works before. Non-exhaustive examples of some of these works, sorted by year, are: “PAC-MDL Bounds” (Blum and Langford 2003), arXiv:1912.01439, arXiv:2006.13057, arXiv:2105.01747, arXiv:2110.11216, arXiv:2111.05275, arXiv:2203.02474, arXiv:2303.05369. Although some of these works are less directly related as they consider different approaches (and some maybe are more directly related), I believe it is helpful to put this work in this context as well: how does the unifying approach of this work differ from previous attempts? What are the similarities? What are the “limitations” of those works and this one?


**Questions:**

The main ones are mentioned above. Some other minor typos/comments:

-	Although trivial but K and C are not introduced in line 93.
-	In lines 120-121, it is claimed that “the right-hand side can be further upper-bounded in terms of the information divergence $D(P_{Y|X}\| Q_Y)$ using Proposition 1.” Can it be done for any function $f(y)$?
-	Typo in line 141: $\frac{|gen(w,S)|}{\sigma \sqrt{6/n}}$
-	The proof of Theorem 5, or a comment on how it is derived seems to be missing.
-	There are typos in the proof of Proposition 1;  after line 367, the LHS should be $\frac{d\mu}{d\nu}\log(\frac{d\mu}{d\nu}+1)$. A similar typo exists on the first line after 368.
-	The term “f” seems to be missing in the last line of equations after line 379.


**Limitations:**

Mentioned above.

---

> ### Author Rebuttal · Authors · 2023-08-09
>
> Regarding the decorrelation lemma: both our decorrelation lemma and the Donsker-Varadhan lemma use convex conjugate pairs to bound a product or an expectation of a product. The main difference is that, when using the decorrelation lemma, we can work with some functional of the density ratio $d\mu/d\nu$, which is a more primitive object and can be further relaxed into divergence-like quantities.
>
> - In lines 120-121, it can be done for any $f$ satisfying appropriate moment constraints, although Hölder’s inequality will be needed in general before invoking Proposition 1. We remark, though, that this statement should not be taken as a theorem, but as part of an informal discussion illustrating the use of the decorrelation lemma.
> - The proof of Theorem 5 relies essentially on the same argument as the one used to prove Theorem 4. We will clarify this in the final version.
> - Thank you for catching the typos in the proof of Proposition 1.

---

### Official Review · Reviewer_7JF4 · 2023-07-04

**Soundness:** 3 good
**Presentation:** 3 good
**Contribution:** 2 fair
**Rating:** 6
**Confidence:** 3

**Summary:**

This paper describes some steps of the standard formula to obtain generalization error bounds: (i) decoupling of the joint distribution + (ii) chaining. Then, they use this formula contributing mainly on the first front by deriving new decoupling results based on Orlicz $\psi\_p$-norms that can be cast into the standard decoupling results based on the relative entropy.

Additionally, they also include a section describing results based on couplings between the hypothesis' distribution obtained with the algorithm's kernel evaluated on a training set and the hypothesis' marginal distribution. These are used to develop their bounds on chaining, but they can stand alone independently.

Finally, the authors note that their results can be extended to high-probability PAC-Bayes bounds and that these can be used to also recover bounds of the type of Fernique's and Talagrand's bounds on the expected supremum of a stochastic process.

**Strengths:**

The paper is generally well-written and easy to follow. It does a good work summarizing how to derive generalization error bounds using Orlicz norms and, therefore, using the relative entropy of the posterior hypothesis distribution $P\_{W|S}$ with respect to a prior $Q\_W$.

Their decoupling result (or "decorrelation lemma") is interesting and they gain clarity and interoperability by relaxing the tighter standard result they could have obtained with Young's inequality. I believe that Lemma 1 is the main result of the paper, which is later interpreted and contextualized into the generalization error set-up.

It is good that the presented bounds can be brought down (up to constants) to known bounds in the community. This is usually a necessary "goodness test" to know if the results obtained are valuable. Particularly, it was good to see that Fernique's and Talagrand's bounds were recoverable this way.

[*Even though the strengths section will be shorter than the weaknesses, I believe this is a good paper. The authors should note that this will be the case only to help them improve the manuscript and to expand on each of the weaknesses so it can be easily addressed.*]

**Weaknesses:**

While their "decorrelation lemma" is novel and interesting, the applications of this lemma afterwards for the estimates using couplings, the usage of chaining in the space of measures, and the tail estimates are standard and shown otherwise with different decoupling lemmas (see e.g. [7,9,11,15,16,17,18] of the references of their own paper). While probably not on purpose, the paper seems to position itself as if it is introducing this set of steps to obtain this kind of bounds. This is not accurate, as these procedures have been used and done previously, just with a different (although maybe less general in some situations) decoupling lemma. I believe it would benefit the paper to make this clear both in the introduction and at the start of each of these sections.

The paper obtains results based on the inverse of an Orlicz norm of the Radon-Nikodym derivative of the posterior with respect to the prior. While this seems to be more general, the paper only gives examples of this instantiating into relative entropy-based bounds. In the end, it does not provide any example of anything new to gain by using their framework with respect to what was already known. This is not super problematic, but when the constants of the bounds are deteriorating under this framework, it is good to show that it can produce new results that lead to interesting conclusions.

Similarly to the comment above, the paper does not give examples or motivation to many of their bounds. For instance:
- Why are the bounds using couplings useful? The paper mentions the fact that when the algorithm ignores the data, i.e. $P\_{W|S} = P\_W$ a.s. and the prior $Q\_W$ is chosen to be the real marginal of the hypothesis $P\_W$, then the resulting bound is exactly zero, which is not the case for the bounds presented in Section 4. However, the bounds in Section for are for the expected *absolute* generalization error, where the extra term $O(1/\sqrt{n})$ is hard to avoid (or unavoidable in many cases). In Section 5, the bounds are for the expected generalization error instead, where the standard bounds (e.g. from [7]) already achieve a zero generalization error with this set-up.
- Why are the bound using chaining useful? The paper basically uses the same techniques as [17] with their "decorrelation lemma". In [16,17] they come up with a slightly artificial example to showcase why their results are useful when compared to previous bounds. It is unclear if this technique provides any practical advantage to previous results, either in terms of obtaining bounds with a better rate in certain settings, in terms of interpreting the resulting bounds, or even in terms of simplicity to obtain actually computable bounds.

The paper does not discuss some relevant literature in certain parts of the text:
-  In line 102 they state that they "define the conditional divergence [...]". This is a standard notation for the conditional divergence, sometimes credited to Verdú, although I am unsure which is the origin. Probably this is just a writing thing, but it contrasts with the previous introduction of the relative entropy where they introduce it with "[...] is defined as".
- Even though the techniques employed are different, some mention to [A] would be interesting in the introduction and/or in the preliminaries. In this paper, they obtain PAC-Bayes bounds considering Orlicz norms with general Orlicz function, not necessarily $\exp(x^p) - 1$.
- In Section 5, it would be interesting to compare and/or mention other works that deal with the couplings of the posterior and the prior to bound the generalization error, e.g. [B,C,D]. In [C], for instance, they also mention the relationship of these bounds with chaining in Appendix A and with the relative entropy and subgaussian conditions in Appendix B.
- In Section 7, it would be good to compare and discuss the work in [E] regarding chained generalization error bounds.

**Additional References**

[A] Amedeo Roberto Esposito, Michael Gastpar, and Ibrahim Issa. "Generalization Error Bounds Via Rényi, f-Divergences and Maximal Leakage". IEEE Transactions on Information Theory. 2021.

[B] Hao Wang, Mario Diaz, José Cândido S. Santos Filho, and Flavio P. Calmon. "An Information-Theoretic View of Generalization via Wasserstein Distance". IEEE ISIT. 2019.

[C] Borja Rodríguez-Gálvez, Germán Bassi, Ragnar Thobaben, and Mikael Skoglund. "Tighter Expected Generalization Error Bounds via
Wasserstein Distance". NeurIPS. 2021.

[D] Ron Amit, Baruch Epstein, Shay Moran, and Ron Meir. "Integral Probability Metrics PAC-Bayes Bounds". NeurIPS. 2022.

[E] Eugenio Clerico, Amitis Shidani, George Deligiannidis, and Arnaud Doucet. "Chained Generalisation Bounds". COLT. 2022.



**Questions:**

- I believe in the equations after line 379 there is an $f$ missing in the last equality.

- I don't really follow the set of inequalities after line 381. Could you please clarify it?

- Can (how does) this framework incorporate other advances in generalization error bounds based on mutual information like the single sample bounds from [8], [F], [G] or the data-dependent bounds from [H], [12], [F]?

- How can we interpret the provided bounds? Can new results be obtained from them other than recovering the standard relative entropy-based bounds? Could you give some examples where the abstraction to Orlicz norms is beneficial to the analysis?

**Additional References**

[F] Borja Rodríguez-Gálvez, Germán Bassi, Ragnar Thobaben, and Mikael Skoglund. "On Random Subset Generalization Error Bounds and the Stochastic Gradient Langevin Dynamics Algorithm". ITW. 2020.

[G] Ruida Zhou, Chao Tian, and Tie Liu. "Individually Conditional Individual Mutual Information Bound on Generalization Error". IEEE Transactions on Information Theory. 2022.

[H] Jeffrey Negrea, Mahdi Haghifam, Gintare Karolina Dziugaite, Ashish Khisti, and Daniel M Roy. "Information-theoretic generalization bounds for SGLD via data-dependent estimates". NeurIPS. 2019.

**Limitations:**

The limitations of the paper are not generally pointed out in the paper (e.g. some of the questions and weaknesses above are not discussed such as how the extension to Orlicz spaces gives any advantage to the standard relative entropy and the decoupling lemma using Donsker-Vardhan).

Regarding potential negative societal impact, there is no mention but there is no need to be as this is theoretical, fundamental research.

---

> ### Author Rebuttal · Authors · 2023-08-09
>
> It is not the intent of our work to cover all existing generalization error bounds or even to replace existing approaches (many of which, as this review correctly points out, make use of various decoupling lemmas). Most of the additional references listed in the review are indeed very relevant, and we will cite and contextualize them in the final version.
>
> Our decorrelation lemma can be considered as an alternative to the Donsker-Vardhan lemma that allows us to work directly with the tails of Radon-Nikodym derivatives (or density ratios), rather than with various expected values, such as the relative entropy. Both the $\psi_2$ function and the relative entropy arise naturally whenever one can engineer a subgaussian condition for a properly chosen centered process, say, by symmetrization techniques. This is often possible with minimal assumptions. However, under other tail conditions, it should possible to use a similar approach based on estimating Legendre-Fenchel conjugates to come up with different decorrelation lemmas to cover more general scenarios.
>
> The benefit of using couplings is that we can actually relate the quantities on the upper bounds to an optimal transportation problem. Specifically, a common quantity in our bounds that rely on couplings is of the form ${\bf E}[C(U,V)R(\mu,\nu)] + {\bf E}[C’(\bar{U},\bar{V})]$, where $(U,V),(\bar{U},\bar{V})$ are random pairs in the hypothesis space; $C, C’$ are some cost functions, and $R$ is some functional of the density $d\mu/d\nu$; and $\mu$ is data dependent, while $\nu$ is not. Observe that the second term is the standard transportation cost under the law of $(\bar{U},\bar{V})$, while the first term can be thought of as information-weighted transportation cost due to the presence of $R(\mu,\nu)$, especially if we take $\nu$ to a marginal of $\mu$ (after averaging out the data). Thus, we can interpret $R$ as a functional of the ``information gain" due to observing the data. We did not mention this perspective in the paper because the cost function would be somewhat exotic if we were view it literally in the context of a transportation problem. However, we do mention that the choices of couplings and priors are flexible, such that we may find an optimal coupling and prior for some particular cases.
>
> Now we briefly illustrate the benefit of using chaining in the space of measures. We consider the term ${\bf E}[C(U,V)R(\mu,\nu)]$. When the posterior $\mu$ is not absolutely continuous w.r.t. the prior $\nu$, the functional of the density $d\mu/d\nu$ becomes vacuous. However, if we use chaining, then the term containing the density ratio is weighted by a distance-like quantity as long as we make sure that the entire summation (such as the summation on the RHS of Theorem 4) is finite. This is a much weaker assumption than $\mu \ll \nu$. A relevant example is the Dudley entropy integral where we also have this multiplicative form $\epsilon \sqrt{\log N(\epsilon)}$. Here $\log N(\epsilon)$, the epsilon-entropy, will generally become infinite as $\epsilon$ goes to zero, but this will not be an issue as long as the integral is finite.
>
> Some comments and answers to the questions:
>
> - We were certainly not claiming the definition of conditional divergence to be original; it is standard in the information theory literature, going back at least to the text of Csiszár and Körner. We should have been more consistent in using the passive voice (''is defined as" rather than
> ''we define it as").
> - There is indeed a missing $f$ after line 379.
> - Clarification of the chain of inequalities after line 381: the first inequality uses the definition of $E$ to replace $g$; the second inequality uses the fact $1 \le \langle \nu, \exp(g^p) \rangle$; the third inequality is due to $(a+b)^{1/p} \le a^{1/p} + b^{1/p}$ for $a, b \ge 0$.
> - Partial answers to the last two questions are addressed in the previous paragraphs.

---

> > ### Comment · Reviewer_7JF4 · 2023-08-14
> > **Answer to rebuttal**
> >
> > Thank you for your rebuttals (both the individual one and the general one). Let me expand further on some things that remain unclear to me.
> >
> > 1. You mention that the decorrelation lemma allows you to work with the tails of the Radon-Nikodym instead of various expected values, such as the relative entropy. It is true that $D(\mu \Vert \nu) = \mathbb{E}\_\mu \log \frac{d\mu}{d\nu}$, but you also deal with $\mathbb{E}\_\mu f \psi\_p^{-1} \frac{d\mu}{d\nu}$. Hence, I do not agree with that statement.
> >
> > 2. Continuing with the point above, I agree that working with an Orlitz norm gives you more generality than working with the relative entropy and that a better understanding of the tail of $\frac{d\mu}{d\nu}$ can be gained in certain situations. However, I am still missing some examples of this. Could you give some example of some "common" tail behavior where the abstraction to an Orlitz norm provides some advantage? I believe this would improve the paper substantially.
> >
> > 3. Regarding the couplings. I like this interpretation, I believe it should be mentioned in the paper to give some intuition. I still do not see there is a practical benefit on its own, but I like it conceptually. Also, I enjoyed that you can use it together with chaining to obtain Wasserstein-2 based bounds without a Lipschitz assumption. Could that be done without the need of chaining?
> >     * I still believe that in Section 5 it should be acknowledged that the fact that bounds that go to 0 can be obtained is not from the fact that one is using couplings, but to the fact that one is bounding the generalization error instead of the absolute generalization error.
> >
> > 4. Regarding chaining. Again, I like the bound presented in the general rebuttal, it is conceptually nice. I understand that the successive refinements will make the bound not to go to infinity in certain situations. Also, I appreciate Dudley's metric entropy to bound the suprema of processes. \
> > However, we have applications of the metric entropy to real scenarios. For instance, we can study the Wasserstein concentration of the empirical measure to the true measure, and obtain a true rate of $\mathcal{O}(1 / \sqrt{n})$. On the other hand, I do not see how this can be used for the generalization error of a learning algorithm. The examples given in [16,17] are quite artificial. Could you give some example where we can see some application of this bound to a learning problem?

---

> > > ### Author Response · Authors · 2023-08-14
> > > **Thank you for the additional questions.**
> > >
> > > 1. The presence of the nonnegative function $f$ allows for incorporating information about the tail behavior of $d\mu/d\nu$, e.g., we could take $f = {\bf 1}_{\lbrace d\mu/d\nu \ge e^r \rbrace}$ for some threshold $r$, or we could use some other indicator of a tail event. The expectation ${\bf E}[f \psi^{-1}_p(d\mu/d\nu)]$ could be further upper-bounded using Hölder's inequality. We have not had the occasion to use arguments of this form in the paper, but we interpreted the original question about the decorrelation lemma vs. Donsker-Varadhan in general terms, not just in the context of our work. Moreover, the tail estimates in Section 7 are an illustration of how the decorrelation lemma can be used beyond expectation values (cf. the first paragraph of Section 7).
> > >
> > > 2. Given the space and time constraints, we don't think we could come up with an example right now that would not be contrived in some sense. Overall, as we have emphasized in our original rebuttal, we do not subscribe to the idea that every new paper on generalization bounds is in ''competition'' with earlier work and thus has to demonstrate some ''advantage'' in terms of the concrete examples it can handle that previous work could not. We do, however, believe that our approach has certain conceptual advantages over some of the earlier work; in particular, many of the existing results that have relied on tailor-made constructions can now be explained in a more transparent way.
> > >
> > > 3. The benefit of the Wasserstein-2  bounds is that they can be combined with chaining. By letting $K=1$, we have one term instead of summation of several term.
> > > - Sorry about the confusion. You are right that the reason we can obtain $0$ for the example in Sec. 5 is that we are bounding the generalization error instead of the absolute generalization error. However, what we would like to emphasize is that using coupling can remedy the issue of the extra term that arises from our use of the decorrelation lemma instead o decoupling using Donsker-Varadhan. For example, in Theorem 1, without coupling the best thing we can do to bound the generalization error is to first bound it with the absolute generalization error, in which case a term one the order of $1/\sqrt{n}$ will always be present.
> > >
> > > 4. Since computing or estimating covering numbers is a much easier task, Dudley’s entropy integral is usually more practical to use. However, the Talagrand-Fernique bound on the suprema of subgaussian processes in terms of majorizing measure is sharp, unlike Dudley’s entropy integral. Our bound treats the supremum as expectation with respect to a random measure and thus recovers the Talagrand-Fernique majorizing measure bound. The key idea here is that we can derive both the bounds of [16] and [17] and the (sharp) Talagrand-Fernique bound using our methodology. This suggests that we can handle less artificial situations than the ones considered in [16] and [17] by building on our approach together with that of [18]. Again, given the constraints of space and time, we feel that this is better left for future work.

---

> > > > ### Comment · Reviewer_7JF4 · 2023-08-15
> > > > **Thanks for the clarifications**
> > > >
> > > > Thank you for your clarifications. Everything is resolved. Let me just expand on each your explanations.
> > > >
> > > > 1. Of course, I missed that! I see, thanks!
> > > >
> > > > 2. I don't think that every new paper on generalization bounds is in "competition" with earlier work and needs to demonstrate some "advantage" in terms of concrete examples. I believe there is value in proposing a new framework with different tools and interpretations, as I already mentioned in the original review. However, I do believe that if, when introducing a new, more general framework, one has examples of common behaviors where the abstraction provides an advantage is better than if it does not. I hope this can be done in a journal extension of the paper or in future work.
> > > >
> > > > 3. Right, so this means that we can choose a $\rho\_{W\_1, W\_0}$ such that $$\frac{1}{\sqrt{n}} \bigg( \mathbf{E}[W\_2 (P\_{W|S}, Q\_W)] + \mathbf{E} \Big[ W\_2 (P\_{W|S}, Q\_W) \sqrt{ D(P\_{W,W'|S} \Vert P\_{W,W'})  } \Big] \bigg) = \frac{1}{\sqrt{n}} \bigg( \mathbf{E}[W\_2 (P\_{W|S}, Q\_W)] + \mathbf{E} \Big[ W\_2 (P\_{W|S}, Q\_W) \sqrt{ D(P\_{W|S} \Vert Q\_{W})  } \Big] \bigg)$$
> > > > where I let $W' \sim Q\_W$?  \
> > > > In any case, I would like to see the example from the general rebuttal and its particularization to $K=1$ in terms of $P\_{W|S}$ and the prior $Q\_W$ (maybe the equation above if I am not mistaken) in the final manuscript if the extra page allows it.
> > > >
> > > > * Thanks, I understand it now. Could you please clarify that a little more in the main text for the final version?
> > > >
> > > > 4. I understand. As for 2. above, I hope this can be done in a journal extension of the paper of in future work.

---

> > > > > ### Author Response · Authors · 2023-08-18
> > > > > **These are all excellent suggestions!**
> > > > >
> > > > > Thank you for the careful reading of the paper and for probing further. This discussion has been very helpful, and we will definitely incorporate your suggestions in the final version.

---

### Official Review · Reviewer_U1pj · 2023-07-05

**Soundness:** 3 good
**Presentation:** 3 good
**Contribution:** 3 good
**Rating:** 7
**Confidence:** 3

**Summary:**

The paper presents a unified perspective of information-theoretic generalization bounds through decorrelation and coupling/chaining. Various existing generalization bounds in the literature are recovered or generalized via this perspective. The Fernique-Talagrand upper bound on the expected supremum of subgaussian processes also emerge as a special case from this framework.


**Strengths:**


I did not check carefully the proofs in the supplementary materials, but the presented results look sound to this reviewer. The paper offers a good service to the learning theory community for understanding information-theoretic generalization bounds and key techniques therein for their development. The development in this paper is original, to this reviewer.

The paper is also clearly written. I appreciate the authors' informal summary in section 1.1, making the paper easy to read.



**Weaknesses:**

On the minor side, the work appears to be limited only to unification, without aiming at developing novel and/or tighter bounds using the developed tool, despite that some bounds are stated in more general forms (e.g., Theorem 7) and some are shown to have certain improvement (e.g. Theorem 3). It would be much better appreciated if the authors can further demonstrate the power of their framework by presenting novel bounds of greater significance.

**Questions:**

In the discussion after Theorem 3, the authors demonstrated the advantage of the theorem in the trivial case where W and S are independent. Are there more interesting (i.e. less trivial) cases where such an advantage can be demonstrated?

**Limitations:**

See weakness.

---

> ### Author Rebuttal · Authors · 2023-08-09
>
> For Theorem 3: if the output of the algorithm does depend on the data, we have to choose suitable couplings and prior, such that the sum of the two terms on the right-hand side of (12) is minimized. We doubt that there exists a single procedure for finding such optimal choices in general cases (or even in ``less trivial cases''), but this is definitely an important direction for future work. Also, we use Theorem 3 to obtain results beyond the independent case. For example, via Corollary 3, Theorem 3 leads to the results of Section 6 (Theorem 4 and 5), which make nontrivial use of couplings beyond the case of independent $S$ and $W$.

---

> > ### Comment · Reviewer_U1pj · 2023-08-10
> > **Thank you for the response.**
> >
> > I am keeping my rating.

---

### Official Review · Reviewer_5Qrr · 2023-07-07

**Soundness:** 3 good
**Presentation:** 3 good
**Contribution:** 3 good
**Rating:** 6
**Confidence:** 4

**Summary:**

This paper proposes a unified framework for deriving information-theoretic generalization bounds for learning algorithms. The main technical result relies on a probabilistic decorrelation lemma based on a change of measure and Young’s inequality in $L_{\psi_p}$ Orlicz spaces. Combining it with other techniques, such as summarization, couplings, and changing the space of probability measures, new upper bounds on the generalization error can be obtained both in expectation and in high probability. The proposed framework also recovers many of the existing generalization bounds as special cases, including the ones based on mutual information, conditional mutual information, stochastic chaining, and PAC-Bayes inequalities. Strength: Lemma 1 provides a very general way to derive generalization error bounds in multiple setting, and it recovers many existing results as special case.

**Strengths:**

The decorrelation Lemma 1 provides a very general way to derive generalization error bounds in multiple settings, and it recovers many existing results as a special case.

**Weaknesses:**

As the proposed bounds are not based on standard information measures, is there a way to evaluate these quantities even in some illustrative examples, like the mean estimation problem considered in [8] and [17]?

**Questions:**

Could the proposed framework recover the ICIMI bounds obtained in the following paper? In general, is it compatible with the individual sample technique used in [8] and [12]?

Zhou, Ruida, Chao Tian, and Tie Liu. "Individually conditional individual mutual information bound on generalization error." IEEE Transactions on Information Theory 68, no. 5 (2022): 3304-3316.


**Limitations:**

The limitation has been addressed well in the confusion section.

---

> ### Author Rebuttal · Authors · 2023-08-09
>
> Since our decorrelation lemma can be used as an alternative to Donsker-Varadhan, it should be possible to obtain the ICIMI bounds using our approach since a key lemma used to prove these bounds also makes use of Donsker-Varadhan. Moreover, it should be possible to obtain the results of [8] and [12] proved using the approach of [14], which also relies on Donsker-Varadhan and random subsampling of the training data. The same technique as we used in our proof of the original CMI bound of Zakynthinou and Steinke could then be used to recover the main results of [14] (possibly with different constants).

---

> > ### Comment · Reviewer_5Qrr · 2023-08-21
> > **Thank you for the response.**
> >
> > I believe the paper will benefit from discussing some simpler examples, and I will keep my score.

---

### Author Rebuttal · Authors · 2023-08-09

We would like to thank all the reviewers for their detailed and careful reviews. In fact, we wish all of NeurIPS reviews adhered to such a high standard! While we address the points raised by each reviewer in individual rebuttals, we would like to clarify a few points that were common to all the reviews.

The literature on information-theoretic generalization bounds is rather extensive and continues to grow. We thank the reviewers for mentioning a number of works we should have cited; we will do so in the final version and discuss their relation to other work whenever possible. We neither believe nor claim that all existing results can be unified under a single framework. The unification we speak of in our work is to ``effectively combine the information-theoretic approach with the classical framework based on various measures of complexity of the hypothesis class" (lines 29-30 in our paper). This can be (and has been) done in different ways; we believe our approach is valuable because it rests on a couple of simple but very flexible ingredients and because it seamlessly interpolates between classical bounds of Fernique and Talagrand and more recent information-theoretic results.

Also, here we would like to provide an example we did not include in the paper due to space limitations. This example may help to provide more intuition about our results, specifically with respect to the use of couplings as in Theorem 3.

Assume the loss function satisfies (13) as in Theorem 5. We take $Q_W = P_W$, take $P_{W_k|S}$ to be some points on the geodesic with respect to Wasserstein-2 distance with endpoints $P_{W|S}$ and $P_W$, and take $\rho_{W_kW_{k-1}}=P_{W_kW_{k-1}}$. Then there exist data-dependent couplings $P_{W_k W_{k-1}|S}$, such that

$$
{\bf E}[{\rm gen}(W,S)] \lesssim \frac{1}{\sqrt{n}}\left( {\bf E}[W_2(P_{W|S}, P_W)] + \sum_{k=1}^K {\bf E}[W_2(P_{W_k|S}, P_{W_{k-1}|S})\sqrt{D(P_{W_k W_{k-1}| S}|| P_{W_k W_{k-1}} )}]\right).
$$

Observe that the first term is the expected Wasserstein-2 distance between the posterior and the prior and the second term is a sum of  ‘’divergence weighted’’ Wasserstein distances. Also note that the form of the second term is in the spirit of the Dudley entropy integral, where the Wasserstein distance corresponds to the the radius of the covering ball and the square root of the divergence corresponds to square root of the metric entropy. We believe such a bound, where the Wasserstein distance appears without Lipschitz assumption on the loss function, is new. As far as we can tell, it is not covered by references [B], [C], and [E] brought up by Reviewer 7JF4.

---

### Decision · Program_Chairs · 2023-09-21

**Decision:**

Accept (poster)

**Comment:**

The paper offers a unified perspective on information-theoretic generalization bounds by incorporating concepts of decorrelation and coupling/chaining. This perspective allows the recovery and generalization of existing generalization bounds in the literature, as well as showcasing how the Fernique-Talagrand upper bound on subgaussian processes emerges as a special case. The presentation is clear, and while the reviewers haven't extensively verified the proofs in the supplementary materials, the results seem sound. The paper's primary strength lies in its contribution to understanding information-theoretic generalization bounds and the techniques behind them.